# DNA-PK deficiency potentiates cGAS-mediated antiviral innate immunity

Xiaona Sun[1,2,10], Ting Liu[1,2,10], Jun Zhao [3,10], Hansong Xia[3,4], Jun Xie[1,2], Yu Guo[1,2], Li Zhong [2], Mi Li [2], Qing Yang[2], Cheng Peng[5], Isabelle Rouvet[6], Alexandre Belot [7,8,9], Hong-Bing Shu [2], Pinghui Feng [3✉] & Junjie Zhang [1,2✉]

Upon sensing cytosolic DNA, the enzyme cGAS induces innate immune responses that underpin anti-microbial defenses and certain autoimmune diseases. Missense mutations of *PRKDC* encoding the DNA-dependent protein kinase (DNA-PK) catalytic subunit (DNA-PKcs) are associated with autoimmune diseases, yet how DNA-PK deficiency leads to increased immune responses remains poorly understood. In this study, we report that DNA-PK phosphorylates cGAS and suppresses its enzymatic activity. DNA-PK deficiency reduces cGAS phosphorylation and promotes antiviral innate immune responses, thereby potently restricting viral replication. Moreover, cells isolated from DNA-PKcs-deficient mice or patients carrying *PRKDC* missense mutations exhibit an inflammatory gene expression signature. This study provides a rational explanation for the autoimmunity of patients with missense mutations of *PRKDC*, and suggests that cGAS-mediated immune signaling is a potential target for therapeutic interventions.

[1] The State Key Laboratory Breeding Base of Basic Science of Stomatology & Key Laboratory of Oral Biomedicine Ministry of Education, School & Hospital of Stomatology, State Key Laboratory of Virology, Wuhan University, Wuhan, China. [2] Frontier Science Center for Immunology and Metabolism, Medical Research Institute, School of Medicine, Wuhan University, Wuhan, China. [3] Section of Infection and Immunity, Herman Ostrow School of Dentistry, Norris Comprehensive Cancer Center, University of Southern California, Los Angeles, CA, USA. [4] Department of Orthopaedics, 3rd Xiangya Hospital, Central South University, Changsha, China. [5] Department of Burns and Plastic Surgery, 3rd Xiangya Hospital, Central South University, Changsha, China. [6] Hospices Civils de Lyon, Centre de Biotechnologie Cellulaire et Biothèque, Bron, France. [7] Centre International de Recherche en Infectiologie, CIRI, Inserm, U1111, Université Claude Bernard Lyon 1, CNRS, UMR5308, École Normale Supérieure de Lyon, University of Lyon, Lyon, France. [8] National Referee Centre for Pediatric-Onset Rheumatism and Autoimmune Diseases (RAISE), Lyon, France. [9] Hospices Civils de Lyon, Paediatric Nephrology, Rheumatology, Dermatology Unit, Mother and Children University Hospital, Bron, France. [10]These authors contributed equally: Xiaona Sun, Ting Liu, Jun Zhao. ✉email: pinghui.feng@usc.edu; junjiezhang@whu.edu.cn

Innate immunity is the first line of host defense against invading pathogens. In response to infection, cells deploy diverse DNA sensors to detect microbial DNA or aberrantly localized cellular DNA and elicit antiviral immune responses. Among the receptors implicated in sensing DNA, cyclic GMP-AMP (cGAMP) synthase (cGAS) plays a pivotal role in detecting double-stranded DNA (dsDNA) in the cytosol[1]. Upon binding dsDNA, cGAS catalyzes the synthesis of cGAMP from ATP and GTP. cGAMP serves as a second messenger to induce the dimerization and activation of the endoplasmic reticulum-anchored stimulator of interferon gene (STING, also known as MITA)[2,3]. STING then recruits and activates TANK-binding kinase 1 (TBK-1) and IKKβ. These kinases in turn activate interferon regulatory factor 3 (IRF3) and nuclear factor-κB to induce the expression of inflammatory cytokines, type I interferon (IFN) and IFN-stimulated genes[4].

The critical role of cGAS in eliciting antiviral innate immunity has been highlighted in host defense against diverse viruses. DNA viruses, including herpesviruses, vaccinia virus, and adenovirus activate innate immune responses through cGAS[5–9]. Surprisingly, RNA viruses, such as West Nile virus and Dengue virus, also trigger cGAS-mediated antiviral immunity that is dependent on mitochondrial DNA (mtDNA)[10,11]. Since cGAS activation potently suppresses viral replication, viruses have evolved intricate strategies to antagonize cGAS-mediated immune signaling[12]. Indeed, we have reported recently that herpes simplex virus-1 (HSV-1)-encoded UL37 deamidase targets cGAS for deamidation to suppress antiviral innate immunity[13].

In the absence of microbial infection, accumulating evidence indicates that cGAS also senses cellular stress and genomic instability. Mitochondrial stress, micronuclei, and cytosolic chromatin fragments have been shown to activate cGAS and induce inflammatory responses[14–16]. While the sensing of viral DNA by cGAS promotes beneficial antiviral responses, the detection of self-DNA and overactivation of the cGAS-STING signaling cascade are associated with autoimmune diseases. For example, gain-of-function mutations of STING as a driver of STING-associated vasculopathy with onset in early infancy have been extensively investigated in both human patients and mouse models[17,18]. These and other studies support the conclusion that the cGAS-STING signaling needs to be tightly regulated to maintain immune homeostasis. In line with this, the enzymatic activity of cGAS is modulated by multiple posttranslational modifications[13,19–24]. Interestingly, it has been reported recently that cGAS localizes to the plasma membrane of macrophages to prevent the recognition of self-DNA[25].

The DNA-dependent protein kinase (DNA-PK) is a serine/threonine protein kinase that is best studied in non-homologous end joining required for double-strand break repair and V(D)J recombination[26]. DNA-PK is a heterotrimeric protein complex consisting of Ku70, Ku80, and the catalytic subunit DNA-PKcs. Besides its nuclear localization, DNA-PK partly localizes in the cytoplasm[27,28] with its cytoplasmic function not well defined. Mutation in the gene encoding DNA-PKcs (PRKDC) leads to severe combined immunodeficiency (SCID) in patients and mice due to V(D)J recombination defect[29–31]. Surprisingly, most patients with DNA-PKcs mutations also suffer from autoimmune diseases due to overactivated innate immunity, which cannot be explained by any known function of DNA-PK[32,33].

In this study, we report that pharmacological and genetic inhibition of DNA-PK promotes cGAS-mediated antiviral innate immunity. Mechanistically, we found that DNA-PK phosphorylates cGAS and suppresses cGAMP synthesis. Treatment with a DNA-PK inhibitor in mice leads to enhanced antiviral immunity and promotes viral clearance. Furthermore, missense mutations of DNA-PKcs in patients lead to enhanced innate immune activation at basal level and upon viral challenge. These findings identify a strategy of immune homeostasis maintenance, whereby cGAS phosphorylation by DNA-PK functions as a checkpoint to restrict overactivated immune responses. Our study also provides a rational explanation for the autoimmunity phenotype observed in patients with PRKDC mutations, suggesting that the cGAS-mediated immune signaling is a potential target for therapeutic interventions.

## Results

**DNA-PK deficiency restricts the replication of vesicular stomatitis virus (VSV) and HSV-1.** To screen for compounds with antiviral activity, we treated THP-1 cells with a library of compounds and then infected them with a prototype RNA virus VSV carrying a luciferase reporter. As a positive control, Ribavirin dramatically inhibited VSV replication. Remarkably, Nu7441, a specific inhibitor for DNA-PK, showed a strong anti-VSV activity (Fig. 1a). To validate this result, we infected THP-1 cells with VSV-GFP (green fluorescent protein) and then quantified viral infection by flow cytometry. Nu7441 treatment strongly suppressed GFP expression in a dose-dependent manner (Fig. 1b and Supplementary Fig. 1a). We also quantified the viral titer in the medium 16 and 24 h post-infection and further confirmed that Nu7441 treatment efficiently reduced VSV replication in both THP-1 and L929 murine fibroblast cells (Supplementary Fig. 1b, c). Next, we examined whether HSV-1, a model of DNA virus, could be inhibited by Nu7441 treatment. Similarly, Nu7441 treatment suppressed the replication of HSV-1 in THP-1 and human foreskin fibroblast (HFF) cells (Fig. 1c, d).

Nu7441 is a highly specific inhibitor of DNA-PK, thus we reasoned that inhibition of DNA-PK activity enhances antiviral activity against both VSV and HSV-1. To eliminate the potential off-target effects of Nu7441, we knocked down DNA-PKcs (Fig. 1e), the catalytic subunit of DNA-PK, and confirmed that DNA-PKcs deficiency potently suppressed the replication of both VSV and HSV-1 (Fig. 1f, g). Ku70 and Ku80, the regulatory submits of DNA-PK, form a heterodimer and are required to stabilize each other. Knocking down Ku70 or Ku80 indeed led to destabilization of the other and showed significant antiviral activity against VSV and HSV-1 (Supplementary Fig. 1d, e). These results collectively support the conclusion that DNA-PK deficiency restricts the replication of VSV and HSV-1.

**DNA-PK deficiency promotes antiviral innate immunity.** Since DNA-PK deficiency suppresses the replication of both VSV and HSV-1, we reasoned that inhibition of DNA-PK may enhance antiviral innate immunity. Indeed, Nu7441 treatment significantly increased the expression of IFNB1 and CXCL10 induced by HSV-1 and VSV in both monocytes and fibroblasts (Fig. 2a, b and supplementary Fig. 2a). Immunoblotting analyses further showed that Nu7441 treatment increased the phosphorylation of TBK1 and IRF3 induced by VSV and HSV-1 (Fig. 2c, d). Nu7026, as another widely used DNA-PK inhibitor, also efficiently increased the expression of innate immune genes induced by HSV-1 and VSV (Fig. 2e, f). Notably, both HSV-1 and VSV infection led to the activation of DNA-PK (phospho-DNA-PKcs S2056 as an activation marker), which was abolished by Nu7441 treatment, indicating that Nu7441 efficiently inhibits the activity of DNA-PK (Supplementary Fig. 2b). To demonstrate the specificity of Nu7441, we treated DNA-PKcs knockdown stable cells with Nu7441 and found that the treatment failed to enhance the induction of IFNB1 and CXCL10 upon HSV-1 infection (Supplementary Fig. 2c).

Consistently with the inhibitor results, knockdown of DNA-PKcs led to enhanced expression of IFNB1 and CXCL10 induced

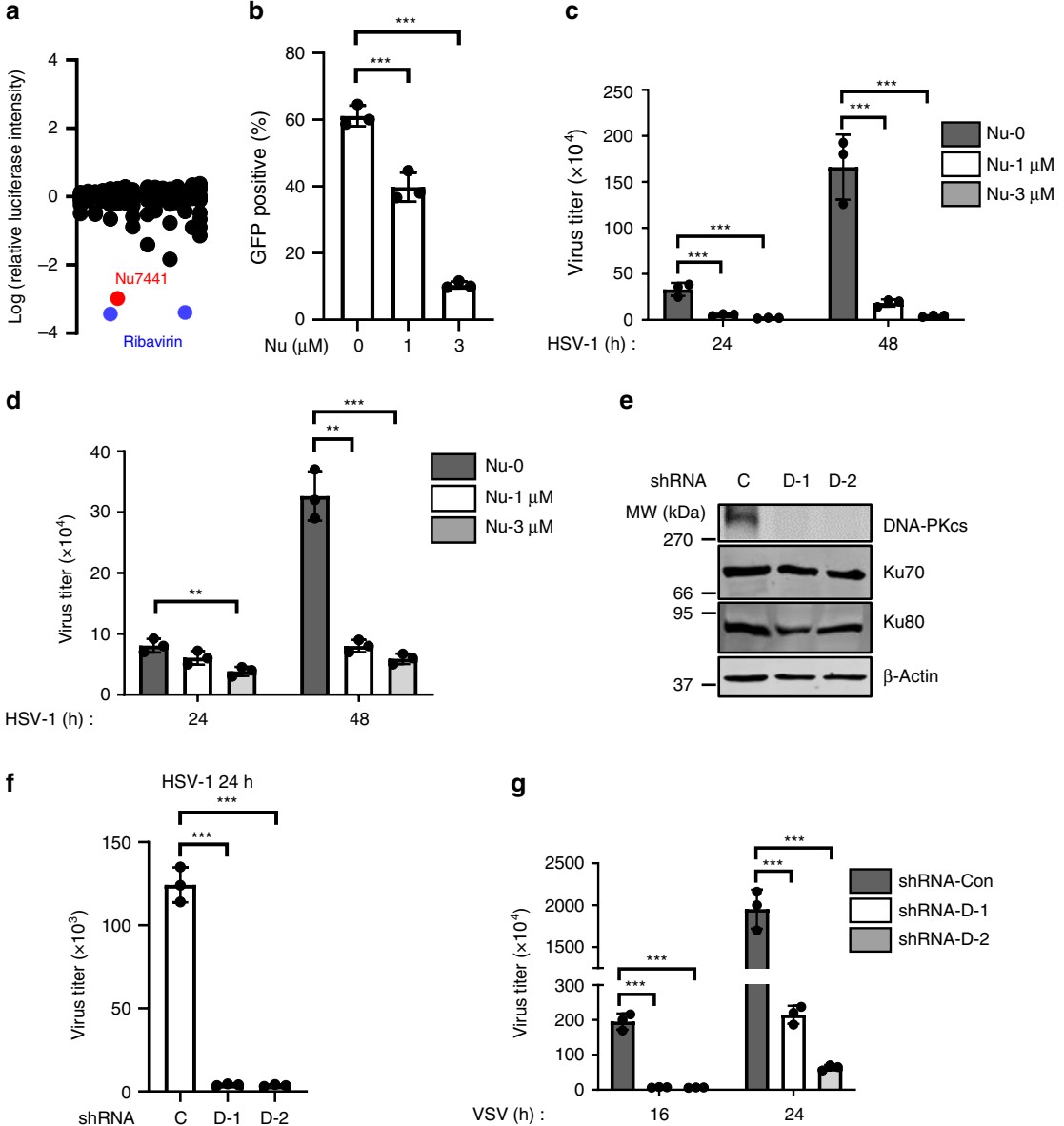

**Fig. 1 DNA-PK inhibition suppresses VSV and HSV-1 replication. a** THP-1 cells were treated with ~150 compounds (10 µM) and infected with VSV-luciferase (MOI = 0.01). Luciferase activity was measured 16 h post-infection and the values were normalized to solvent control. **b** THP-1 cells were treated with Nu7441 and infected with VSV-GFP (MOI = 0.01). GFP-positive cell percentage was quantified by flow cytometry at 16 h post-infection. $p = 0.0022$ (Nu = 1 µM); $p = 1 \times 10^{-5}$ (Nu = 3 µM). **c**, **d** THP-1 or HFF cells were treated with Nu7441 and infected with HSV-1 (MOI = 0.01). Viral titer in the medium at the indicated time points was determined by plaque assays. **c** HSV-1 (24 h): $p = 0.0025$ (Nu = 1 µM), $p = 0.0015$ (Nu = 3 µM); HSV-1 (48 h): $p = 0.0020$ (Nu = 1 µM), $p = 0.0014$ (Nu = 3 µM). **d** HSV-1 (24 h): $p = 0.0054$ (Nu = 3 µM); HSV-1 (48 h): $p = 0.0005$ (Nu = 1 µM), $p = 0.0004$ (Nu = 3 µM). **e** THP-1 cells infected with control (C) or DNA-PKcs (D-1, D-2) shRNA lentivirus were selected with puromycin, and whole-cell lysates (WCLs) were analyzed by immunoblotting with the indicated antibodies. **f**, **g** THP-1 stable cells as described in **e** were infected with HSV-1 (**f**) or VSV (**g**) (MOI = 0.01). Viral titer in the medium at the indicated time points was determined by plaque assays. **f** $p = 4 \times 10^{-5}$ (D-1), $p = 4 \times 10^{-5}$ (D-2); **g** VSV (16 h): $p = 0.0002$ (D-1), $p = 0.0002$ (D-2); VSV (24 h): $p = 0.0002$ (D-1), $p = 0.0002$ (D-2). All experiments were done at least twice, and one representative is shown. $n = 3$ biologically independent samples for **b**–**d**, **f**, **g**. Data are presented as mean values ± SD. \*\*$p < 0.01$, \*\*\*$p < 0.005$, two-tailed Student's $t$ test. Source data are provided as a Source data file.

by HSV-1 and VSV (Fig. 2g, h). Moreover, knockdown of DNA-PKcs in THP-1 cells triggers a more robust activation of the IFN signaling, as evidenced by the increased phosphorylation of TBK1 and IRF3 after VSV and HSV-1 infection compared to the control THP-1 cells (Supplementary Fig. 2d). Furthermore, THP-1 cells depleted of DNA-PKcs induced significantly higher IFN-β secretion than control cells when infected by VSV (Supplementary Fig. 2e). Next, we examined whether the regulatory components of DNA-PK, Ku70, and Ku80, are involved in antiviral activity.

Knockdown of Ku70 or Ku80 in THP-1 cells led to significantly enhanced expression of *IFNB1* induced by both VSV and HSV-1 (Supplementary Fig. 2f), indicating that Ku70 and Ku80 contribute to DNA-PK-mediated immune suppression. DNA-PK plays a critical role in DNA repair and it is well established that DNA damage leads to the accumulation of cytosolic DNA that activates cGAS[34]. We thus examined whether DNA-PK inhibition or depletion in our experimental settings causes DNA damage. Our results indicated that, while etoposide induces strong DNA

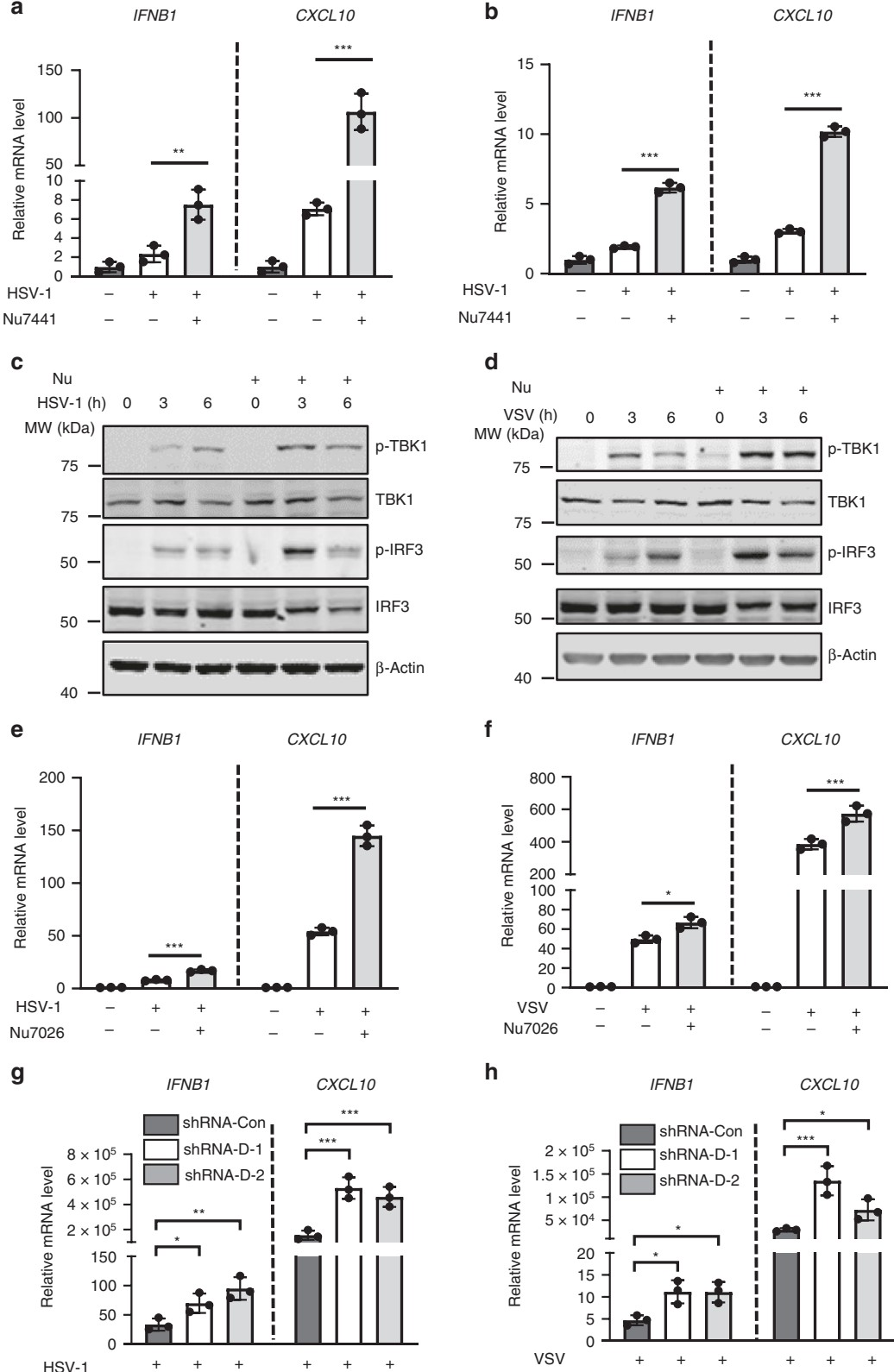

damage response, as evidenced by increased γ-H2AX expression and strong γ-H2AX foci formation, transient knocking down of DNA-PKcs or Nu7441 treatment did not activate detectable DNA damage responses (Supplementary Fig. 2g, h). While DNA damage induced by etoposide elicited innate immune responses, Nu7441 treatment further increased the expression of IFN genes induced by etoposide (Supplementary Fig. 2i). Together, these

data indicate that DNA-PK deficiency promotes cGAS-mediated antiviral innate immunity.

**DNA-PK targets cGAS signaling**. VSV primarily activates the retinoic acid-inducible gene I–mitochondrial antiviral-signaling protein (RIG-I–MAVS) signaling pathway, while HSV-1 engages cGAS-STING signaling to elicit antiviral immune responses.

**Fig. 2 DNA-PK deficiency potentiates stronger innate immune responses. a, b** THP-1 or HFF cells were treated with Nu7441 and infected with HSV-1 (MOI = 5). Infected cells were harvested at 6 hpi, and the expression of the indicated cytokine genes was analyzed by real-time PCR. **a** $p = 0.0076$ (*IFNB1*); $p = 0.0008$ (*CXCL10*); **b** $p = 3 \times 10^{-5}$ (*IFNB1*); $p = 7 \times 10^{-6}$ (*CXCL10*). **c, d** THP-1 cells were mock treated or treated with Nu7441 (3 μM) and infected with HSV-1 or VSV (MOI = 5). Cells were harvested at the indicated time points and whole-cell lysates (WCLs) were analyzed by immunoblotting with the indicated antibodies. **e, f** THP-1 were treated with Nu7026 (10 μM) and infected with HSV-1 or VSV (MOI = 5). Infected cells were harvested at 6 hpi, and the expression of the indicated cytokine genes was analyzed by real-time PCR. **e** $p = 0.0003$ (*IFNB1*); $p = 0.0001$ (*CXCL10*); **f** $p = 0.013$ (*IFNB1*); $p = 0.0049$ (*CXCL10*). **g, h** THP-1 cells stably expressing control or DNA-PKcs shRNA were infected with HSV-1 or VSV (MOI = 5). Infected cells were harvested at 6 hpi, and the expression of the indicated cytokine genes was analyzed by real-time PCR. **g** *IFNB1*: $p = 0.033$ (D-1), $p = 0.0084$ (D-2); *CXCL10*: $p = 0.0023$ (D-1), $p = 0.0039$ (D-2). **h** *IFNB1*: $p = 0.017$ (D-1), $p = 0.013$ (D-2); *CXCL10*: $p = 0.0044$ (D-1), $p = 0.031$ (D-2). All experiments were done at least twice, and one representative is shown. $n = 3$ biologically independent samples for **a, b, e–h**. Data are presented as mean values ± SD. *$p < 0.05$, **$p < 0.01$, ***$p < 0.005$, two-tailed Student's $t$ test. Source data are provided as a Source data file.

Since DNA-PK deficiency suppresses the replication of both VSV and HSV-1, we sought to determine whether these two pathways are regulated by DNA-PK. Employing CRISPR-Cas9 to knock out (KO) cGAS, STING, or MAVS in THP-1 cells (Fig. 3a), we found that cGAS or STING KO nearly abolished the antiviral effect of Nu7441 against VSV and HSV-1, while KO of MAVS had a marginal effect (Fig. 3b, c). These results suggest that DNA-PK inhibits antiviral immune response mediated by cGAS and STING.

Given that RIG-I–MAVS signaling is dispensable for VSV restriction by DNA-PK inhibitor, we sought to determine whether VSV infection activates cGAS. Recently, it has been reported that some RNA viruses, such as West Nile virus and Dengue virus, trigger cGAS-mediated antiviral immunity via DNA released from mitochondria[10,11]. Indeed, VSV, but not Sendai virus (SeV) infection, led to the release of mtDNA into cytoplasm (Fig. 3d and Supplementary Fig. 3a). Consistently, knockdown of DNA-PKcs, Ku70, or Ku80 did not affect the replication of SeV (Supplementary Fig. 3b) nor did it lead to increased innate immune responses in SeV-infected cells (Supplementary Fig. 3c). Moreover, KO of cGAS or STING significantly reduced VSV-induced antiviral innate immune defenses (Supplementary Fig. 3d), confirming that cGAS-STING signaling contributes to the antiviral immunity against VSV.

As Nu7441 treatment did not increase the binding of mtDNA to cGAS in VSV-infected cells, we concluded that DNA-PK does not affect the DNA-binding activity of cGAS (Fig. 3e). Upon sensing DNA, cGAS dimerizes to catalyze the synthesis of cGAMP, which activates downstream signaling. We thus assessed whether DNA-PK inhibition affects cGAS dimerization. VSV but not SeV infection induced cGAS dimerization (Supplementary Fig. 3e). Remarkably, Nu7441 treatment significantly enhanced the dimerization of cGAS (Fig. 3f), suggesting that DNA-PK activation interferes cGAS dimerization.

Herring testes (HT)-DNA is widely used as a specific stimulator of cGAS. Nu7441 inhibition increased the expression of *IFNB1* and *CXCL10* induced by HT-DNA but not poly [I:C] (Fig. 3g, h). In DNA-PKcs knockdown cells, Nu7441 treatment failed to increase HT-DNA-induced innate defenses (Supplementary Fig. 3f). These results further demonstrate that DNA-PK suppresses DNA-induced innate immune activation.

A recent study reports that human DNA-PK activates STING-independent DNA sensing pathway in human cells[35]. We found that HT-DNA-induced innate immune responses are strictly dependent on STING and Nu7441 treatment failed to enhance HT-DNA-induced innate immune responses in STING-KO cells (Supplementary Fig. 3g). HSV-1, on the contrary, induced innate immune responses in STING-KO THP-1 cells. Importantly, Nu7441 treatment reduced HSV-1-induced innate immune responses (Supplementary Fig. 3h), which is consistent with the previous study that DNA-PK positively regulates HSV-1-induced

innate immunity in the absence of STING[35]. In contrast, DNA-PK inhibition by Nu7441 failed to enhance VSV-induced innate immune responses without STING (Supplementary Fig. 3i), which supports our conclusion that cGAS-STING signaling is strictly required for DNA-PK inhibition to restrict VSV replication.

Taken together, these results indicate that DNA-PK targets cGAS-STING signaling to suppress antiviral innate immunity.

**DNA-PK phosphorylates cGAS.** Given that DNA-PK inhibits the cGAS-STING signaling axis and DNA-PK partly localizes in the cytoplasm[27,28], we assessed whether DNA-PK associates with these key signaling components upon virus infection. Co-immunoprecipitation (Co-IP) assay indicated that cGAS interacts with all components of the DNA-PK complex in VSV-infected cells, including DNA-PKcs, Ku70, and Ku80 (Supplementary Fig. 4a). In contrast, STING did not associate with DNA-PK during VSV infection (Supplementary Fig. 4b). Furthermore, endogenous cGAS associates with all components of the DNA-PK complex in HSV-1-infected and HT-DNA-stimulated cells (Fig. 4a and Supplementary Fig. 4c). Collectively, these results demonstrate that DNA-PK interacts specifically with cGAS upon VSV and HSV-1 infection.

Because DNA-PK interacts with cGAS and its kinase activity is required to inhibit cGAS-dependent immune response, we assessed whether DNA-PK can phosphorylate cGAS. In vitro kinase assay indicated that purified DNA-PK phosphorylates cGAS weakly at basal level, while DNA stimulation potently increases the cGAS phosphorylation by DNA-PK (Fig. 4b, c). To determine the sites at which DNA-PK phosphorylates cGAS, we performed in vitro kinase assay by using purified DNA-PK and cGAS with DNA stimulation and analyzed cGAS phosphorylation by tandem mass spectrometry. Two potential cGAS phosphorylation sites (T68 and S213) were identified. We then tested whether phosphorylation on these two sites by DNA-PK would restrict the activity of cGAS. A reporter assay-based examination of the two potential phosphorylation-mimetic mutants (S>D or T>E) revealed that the T68E mutation reduced cGAS-induced activity of the IFN-β promoter to 40% of the activity induced by WT cGAS, while the S213D mutation reduced it to ~15% of that induced by WT cGAS (Fig. 4d). T68E/S213D double mutation further decreased the induction of the IFN-β promoter (Fig. 4d). In contrast, T68A phosphorylation-resistant mutation of cGAS did not affect the IFN-β promoter activity, while cGAS mutants including S213A or T68A/S213A showed significantly reduced IFN-β promoter activity (Supplementary Fig. 4d), likely due to the fact that S213 is in close proximity to the enzymatic core and any mutations/modifications on S213 could disrupt the enzymatic activity of cGAS. Collectively, these results suggest that DNA-PK phosphorylates cGAS at T68 and S213 to restrict its activity.

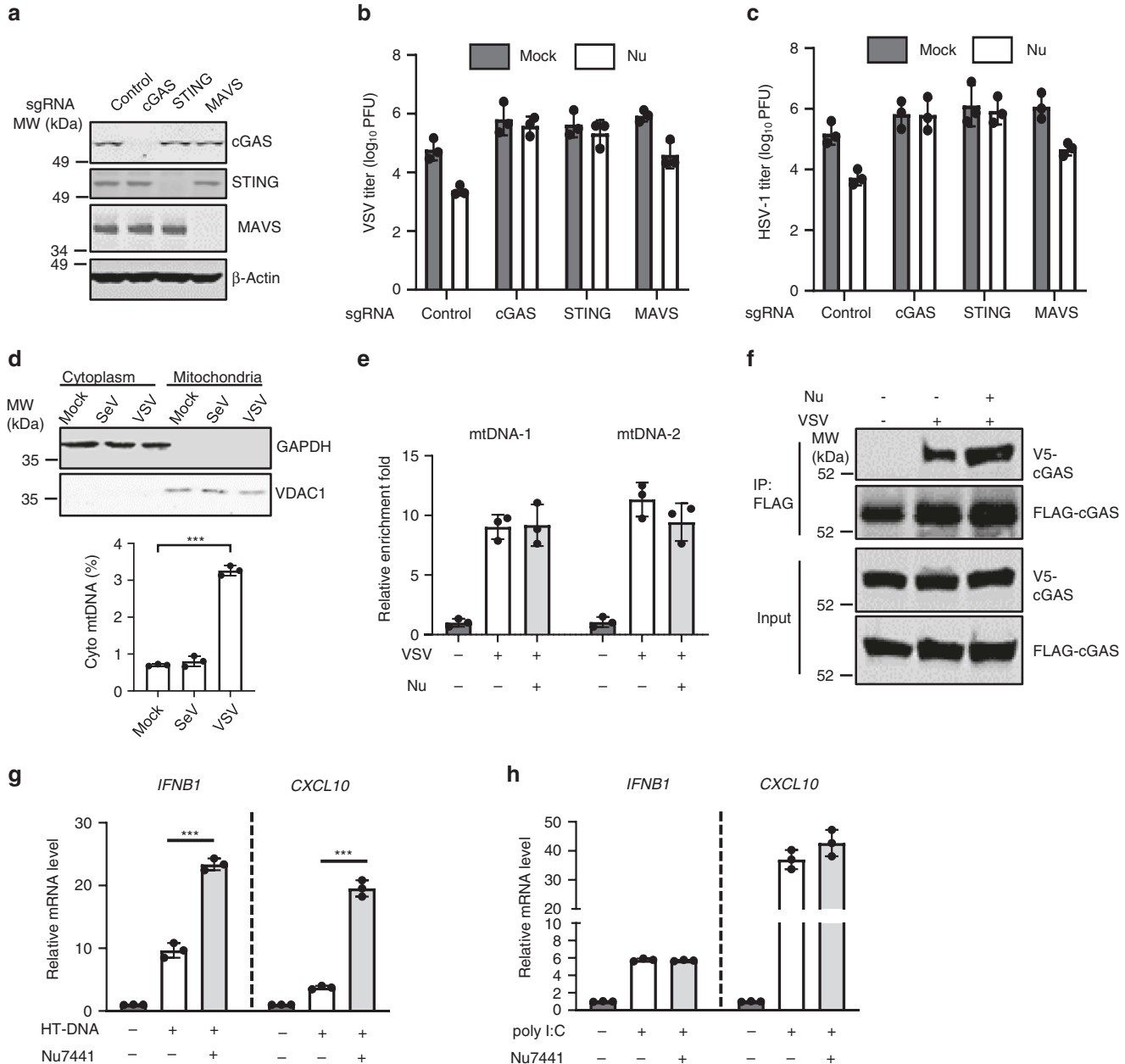

**Fig. 3 DNA-PK inhibition enhances cGAS-mediated antiviral immune responses. a** THP-1 cells were transduced with sgRNA targeting the indicated genes to generate knockout cells. WCLs were analyzed by immunoblotting with the indicated antibodies. **b, c** THP-1 cells as described in **a** were mock treated or treated with Nu7441 (3 μM) and infected with VSV (**b**) or HSV-1 (**c**) (MOI = 0.01). Viral titer in the medium at 24 hpi was determined by plaque assays. **d** THP-1 cells were infected with SeV (20 HAU/ml) or VSV (MOI = 5). Cytoplasmic and mitochondrial compartments were analyzed with the indicated antibodies and cytoplasmic mitochondrial DNA (mtDNA) was extracted and quantified by real-time PCR at 6 hpi. $p = 5 \times 10^{-6}$. **e** 293T cells stably expressing FLAG-cGAS were mock treated or treated with Nu7441 (3 μM) and then were mock infected or infected with VSV (MOI = 0.01) for 16 h. WCLs were precipitated with anti-FLAG agarose and the precipitated DNA was analyzed by Real-Time PCR with mtDNA-specific primers. **f** 293T cells stably expressing FLAG-cGAS and V5-cGAS were mock treated or treated with Nu7441 (3 μM) and then were mock infected or infected with VSV (MOI = 0.01) for 16 h. WCLs were precipitated with anti-FLAG agarose. Precipitated proteins and WCLs were analyzed by immunoblotting with the indicated antibodies. **g, h** THP-1 cells were treated with Nu7441 (3 μM) and transfected with HT-DNA or poly I:C (1 μg/ml). The expression of the indicated genes was analyzed by real-time PCR. **g** *IFNB1*: $p = 9 \times 10^{-5}$, *CXCL10*: $p = 3 \times 10^{-5}$. All experiments were done at least twice, and one representative is shown. $n = 3$ biologically independent samples for **b–e**, **g**, **h**. Data are presented as mean values ± SD. Center and error bars denote mean and SD. ***$p < 0.005$, two-tailed Student's *t* test. Source data are provided as a Source data file.

In vitro kinase assay indicated that wild-type (WT) cGAS was significantly phosphorylated by DNA-PK, while the phosphorylation of T68E and S213D mutants was greatly reduced (Supplementary Fig. 4e). Double mutations of T68E and S213D further decreased cGAS phosphorylation by DNA-PK (Supplementary Fig. 4e). These results indicate that T68 and S213 of cGAS are the major DNA-PK phosphorylation sites in vitro. Next, we generated antibodies specifically recognizing phosphorylated T68 and S213 of cGAS that demonstrated exquisite specificity (Supplementary Fig. 4f). Using these phospho-specific antibodies, we showed that cGAS was phosphorylated at T68 and S213 upon VSV or HSV-1 infection (Fig. 4e and Supplementary

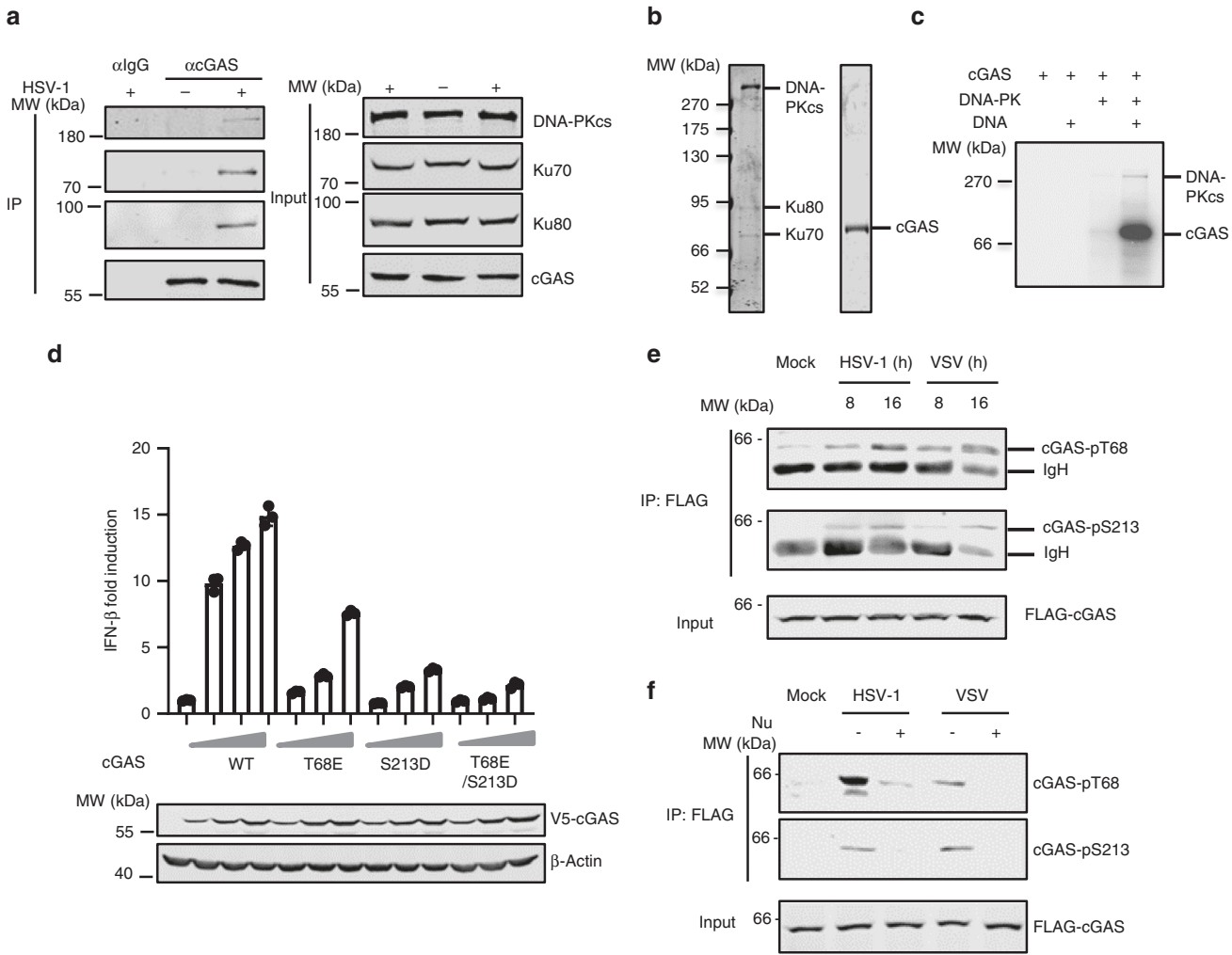

**Fig. 4 DNA-PK phosphorylates cGAS. a** THP-1 cells were mock infected or infected with HSV-1 (MOI = 0.01) for 16 h before co-immunoprecipitation and immunoblotting analysis. **b** DNA-PK complex and purified cGAS were analyzed by silver staining. **c** In vitro kinase assay was performed using DNA-PK complex and purified cGAS with or without HT-DNA (1 μg/ml). **d** 293T cells were transfected with an IFN-β reporter plasmid cocktail with plasmids containing cGAS WT or the mutants and STING. IFN-β activation was determined by luciferase assay. **e** THP-1 cells stably expressing FLAG-cGAS were mock infected or infected with HSV-1 (MOI = 1) or VSV (MOI = 0.1). WCLs were collected at the indicated hpi and precipitated with anti-FLAG agarose. Precipitated proteins and WCLs were analyzed by immunoblotting with the indicated antibodies. **f** THP-1 cells stably expressing FLAG-cGAS were mock infected or infected with HSV-1 (MOI = 1) or VSV (MOI = 0.1) and were mock treated or treated with Nu7441 (3 μM). WCLs were collected at 16 hpi and precipitated with anti-FLAG agarose. Precipitated proteins and WCLs were analyzed by immunoblotting with the indicated antibodies. All experiments were done at least twice, and one representative is shown. n = 3 biologically independent samples for **d**. Data are presented as mean values ± SD. Source data are provided as a Source data file.

Fig. 4g). Remarkably, Nu7441 treatment significantly blunted cGAS phosphorylation at T68 and S213 induced by HSV-1 and VSV (Fig. 4f), supporting the conclusion that cGAS is phosphorylated by DNA-PK at T68 and S213 in virus-infected cells. Together, these results demonstrate that DNA-PK phosphorylates cGAS at T68 and S213 upon VSV and HSV-1 infection to dampen cGAS-mediated innate immune signaling.

**Phosphorylation of cGAS by DNA-PK impairs its oligomerization and enzymatic activity.** To further assess the effect of cGAS phosphorylation by DNA-PK, we reconstituted cGAS$^{-/-}$ L929 cells with WT cGAS or the phospho-mimetic mutants T68E, S213D, or T68E/S213D by lentiviral transduction (Fig. 5a). We observed that the expression of WT cGAS upregulated the expression of *Ifnb*, *Isg56*, and *Cxcl10* at basal level. In stark contrast, the expression of these immune genes in cells reconstituted with cGAS-T68E or -S213D was dramatically reduced,

while cGAS-T68E/S213D failed to induce the expression of these genes (Fig. 5b and Supplementary Fig. 5a). When these cells were stimulated with HT-DNA, we found that cytokine gene expression was significantly reduced in cGAS$^{-/-}$ L929 cells reconstituted with cGAS-T68E, -S213D, or -T68E/S213D compared to that reconstituted with WT cGAS (Fig. 5c and Supplementary Fig. 5b). These results indicate that the phosphorylation of T68 and S213 impairs cGAS-mediated host immune defense toward dsDNA.

Finally, we determined whether cGAS phosphorylation by DNA-PK affects the DNA-binding, dimerizing, and cGAMP-synthesizing activity of cGAS using these phosphorylation-mimetic mutants. We first examined whether the phosphorylation of cGAS by DNA-PK interferes with its dsDNA-binding activity. Biotinylated IFN-stimulating DNA (biotin-ISD) pull-down experiments demonstrated WT cGAS, cGAS-T68E, -213D, and -T68E/S213D interacted with ISD comparably (Supplementary Fig. 5c), consistent with our finding that phosphorylation of

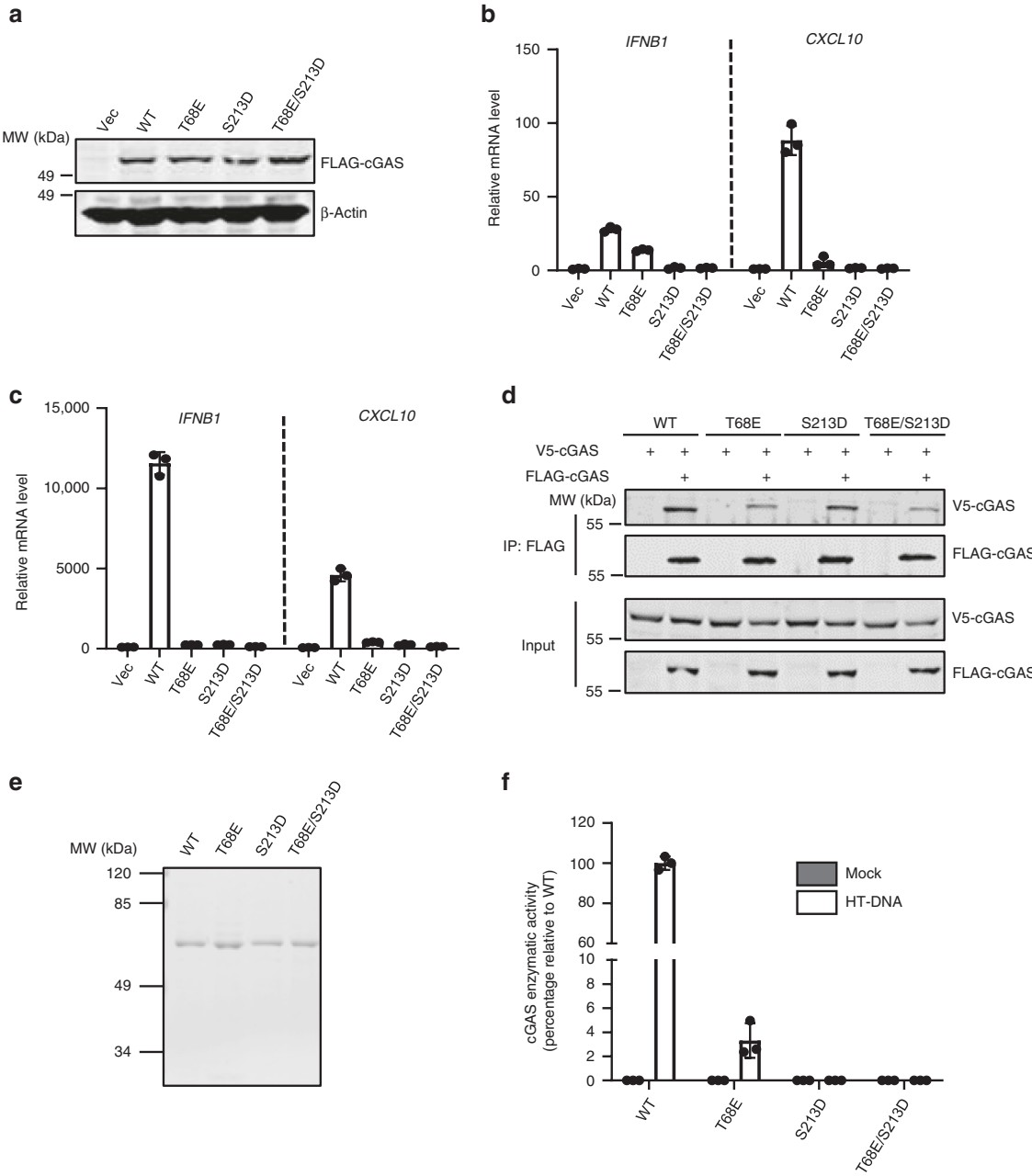

**Fig. 5 cGAS phosphorylation by DNA-PK inhibits the activity of cGAS. a** $cGAS^{-/-}$ L929 cells were infected with control (Vec) lentivirus or that containing cGAS WT or the phosphorylation-mimic mutants. WCLs were analyzed by immunoblotting. **b** Reconstituted cGAS stable cells as described in **a** were selected with puromycin (5 μg/ml) for 3 days. The expression of the indicated inflammatory genes was analyzed by real-time PCR. **c** Reconstituted cGAS stable cells were transfected with HT-DNA (1 μg/ml). Cells were harvested at 6 h post transfection, and the expression of the indicated inflammatory genes was analyzed by real-time PCR. **d** 293T cells were transfected with WT or the mutant cGAS plasmids. WCLs were precipitated with anti-FLAG agarose. Precipitated proteins and WCLs were analyzed by immunoblotting with the indicated antibodies. **e**, **f** Purified cGAS WT or mutants were analyzed by Coomassie blue staining (**e**). In vitro cGAMP activity assay was performed without or with HT-DNA (1 μg/ml) stimulation and the activity was presented as the enzymatic activity percentage against WT cGAS (**f**). All experiments were done at least twice, and one representative is shown. $n = 3$ biologically independent samples for **b**, **c**, **f**. Data are presented as mean values ± SD. Source data are provided as a Source data file.

cGAS by DNA-PK does not impair the dsDNA-binding activity of cGAS. When cGAS dimerization/oligomerization was analyzed by Co-IP assay, T68E and T68E/S213D mutations demonstrated reduced dimerization compared to WT cGAS (Fig. 5d). In contrast, S213D mutation did not significantly affect cGAS dimerization (Fig. 5d). These data are consistent with the previous result that treatment with DNA-PK inhibitor Nu7441 enhances cGAS dimerization. In summary, cGAS phosphorylation by DNA-PK at T68 interferes with its dimerization.

Structurally, S213 is close to the catalytic core of cGAS, which consists of three negatively charged residues (Glu225, Asp227, and Asp319)[10,36]. We reasoned that the phosphorylation of S213 may impinge on the enzymatic activity of cGAS via an electrostatic effect. Indeed, with purified WT cGAS or its mutants, we found that DNA-stimulated cGAMP synthesis activity of the cGAS mutants (T68E, S213D, or T68E/S213D) was dramatically decreased (Fig. 5e, f). Since T68E impairs the dimerization of cGAS that is a prerequisite for cGAS activation, it

is conceivable that T68E mutation interferes with the activation of cGAS. Together, our results show that S213 phosphorylation inhibits the cGAMP synthase activity of cGAS.

Collectively, these results demonstrate that phosphorylation of cGAS by DNA-PK at T68 and S213 distinctly impairs the dimerization and cGAMP synthesis enzymatic activity of cGAS.

**DNA-PK inhibition potentiates antiviral innate immune response in vivo.** Loss-of-function mutation in the gene encoding DNA-PKcs (*PRKDC*) leads to SCID in mice. SCID mice are highly susceptible to viral infection due to the deficiency of T and B lymphocytes, thus excluding the usage of SCID mice to evaluate the antiviral innate immune response. To overcome this challenge, we assessed the effect of DNA-PK-specific inhibitor Nu7441 upon VSV and HSV-1 infection in vivo. Approximately 50% of BL6 mice succumbed to VSV infection with vehicle treatment, while ~90% of the infected mice survived with Nu7441 treatment (Fig. 6a). Consistently, the viral titer of VSV in the spleen of Nu7441-treated mice was reduced by >10-fold compared to that of control mice (Fig. 6b). Moreover, VSV infection led to widespread inflammation and infiltration of immune cells in the lung of the infected mice, which was dramatically alleviated by Nu7441 treatment (Fig. 6c). We also found that Nu7441 treatment induced significantly higher levels of antiviral cytokines (including IFN-β, interleukin (IL)-6, and C-C chemokine motif ligand 5 (CCL5)) in the serum than did vehicle control (Fig. 6d). Next, we determined the effect of Nu7441 treatment on HSV-1 infection. Similarly, while ~30% of BL6 mice survived HSV-1 infection by day 12 post-infection, Nu7441 treatment increased the survival rate to 80% (Fig. 6e). Furthermore, Nu7441 treatment also decreased HSV-1 lytic replication in the mouse brain by ~5-fold, which correlated with significantly enhanced production of antiviral cytokines, including IFN-β, IL-6, and CCL5 (Fig. 6f, g).

To demonstrate the specificity of Nu7441 in vivo, we performed similar infection experiments with cGAS-deficient mice and found that the immune-stimulatory effect of Nu7441 could not be observed (Supplementary Fig. 6a, d). Accordingly, the survival rate of virus-infected mice and viral titers were indistinguishable between the vehicle control and Nu7441 treatment groups (Supplementary Fig. 6b, c, e, f).

Altogether, these results indicate that inhibition of DNA-PK potentiates antiviral innate immunity against both VSV and HSV-1 in vivo.

**DNA-PK deficiency in mouse- and patient-derived cells potentiates antiviral innate immunity.** To further elucidate the role of DNA-PK in regulating innate immunity, we isolated mouse primary lung fibroblasts (MLFs) from SCID and congenic WT mice. Remarkably, we found that MLFs from SCID mice showed significantly higher expression of *Ifnb1*, *Isg56*, and *Cxcl10* at basal level compared to WT MLFs (Fig. 7a and Supplementary Fig. 7a). HSV-1 and VSV infection induced *Ifnb1*, *Isg56*, and *Cxcl10* in WT MLFs, while the expression of these genes was significantly increased in MLFs produced from SCID mice (Fig. 7b and Supplementary Fig. 7b). Similar results were observed with bone marrow-derived macrophages (BMDMs) challenged with HSV-1 and VSV: BMDMs generated from SCID mice expressed higher levels of *Ifnb1*, *Isg56*, and *Cxcl10* compared to those from WT mice (Fig. 7c and Supplementary Fig. 7c). Moreover, the phosphorylation of TBK1 and IRF3 induced by HSV-1 and VSV was enhanced in MLFs derived from SCID mice compared to those from WT mice (Fig. 7d and Supplementary Fig. 7d). When these cells were stimulated with HT-DNA, the expression of *Ifnb1*, *Isg56*, and *Cxcl10* was significantly higher in MLFs and BMDMs from SCID mice compared with those from

WT mice, supporting the conclusion that the activity of DNA-PK is required to dampen cGAS-mediated innate immune response (Fig. 7e and Supplementary Fig. 7e).

Missense mutations of *PRKDC* have been reported in six patients up to 2018[33]. In addition to the expected recurrent infections due to immunodeficiency, one surprising observation is that four out of the six patients suffered from autoimmune diseases, such as granuloma[33]. Our previous results revealed that the activity of DNA-PK is required to restrict cGAS-mediated innate immunity. We aim to assess whether cells from these patients with *PRKDC* mutations have overactivated innate immunity. Indeed, the expression of a panel of IFN-stimulated genes (*IFIT1*, *ISG15*, *SIGLEC1*, *IFI27*, and *RSAD2*) was greatly increased in whole-blood cells of two patients with *PRKDC* mutations (Fig. 7f). Next, we infected fibroblasts generated from control subjects and these two patients with HSV-1 or VSV and examined antiviral innate immune responses. Fibroblasts from patients with *PRKDC* mutations showed significantly higher expression of the antiviral genes, including *IFNB1*, *ISG56*, and *CXCL10* (Fig. 7g and Supplementary Fig. 7f). Conversely, viral replication was greatly reduced in fibroblasts with *PRKDC* mutations compared with control fibroblasts (Fig. 7h and Supplementary Fig. 7g). Together, we conclude that DNA-PK deficiency in mouse- and patient-derived cells potentiates antiviral innate immunity.

## Discussion

By screening for compounds with antiviral activity, we discovered that Nu7441, a specific inhibitor of DNA-PK, potently inhibits the replication of both VSV and HSV-1. We further pinpointed the cGAS–STING signaling axis as the target of DNA-PK and showed that DNA-PK phosphorylates cGAS to suppress innate immune responses. Although VSV has been shown to primarily activate the RIG-I–MAVS signaling axis, we found that VSV infection leads to the release of mtDNA that binds and activates cGAS. These results are consistent with previous studies demonstrating that RNA viruses, such as West Nile virus and Dengue virus, induce antiviral innate immunity through cGAS[10,11]. Furthermore, we found that Nu7441 treatment enhances antiviral innate immunity in vivo and facilitates the clearance of VSV and HSV-1. Remarkably, cells from SCID mice with a loss-of-function mutation in *PRKDC* and patients with *PRKDC* missense mutations have a signature of stronger innate immune and inflammatory gene expression at basal level and in response to viral infection, making them more resistant to viral challenge. Our study thus provides a mechanism whereby the phosphorylation of cGAS by DNA-PK functions as a checkpoint to maintain immune homeostasis.

We identified two sites, i.e., T68 and S213, of cGAS being phosphorylated by DNA-PK. Interestingly, phosphorylation of T68, which locates within the N terminus, interferes with cGAS dimerization, agreeing with a recent report that the N terminus of cGAS is critical for the oligomerization and phase separation of cGAS induced by DNA[37]. It is possible that the phosphorylation of cGAS at T68 by DNA-PK impairs the oligomerization of cGAS and subsequent phase separation. S213, on the other hand, is close to the catalytic core that consists of Glu225, Asp227, and Asp319. Structural studies indicate that S213 is located in the activation loop[38]. As such, our results demonstrate that the phosphorylation of cGAS at S213 dampens the cGAMP synthesis enzymatic activity of cGAS. We reason that the phosphorylation of S213 introduces a negative charge that may impinge on the cGAS-catalyzed cGAMP synthesis. Thus DNA-PK utilizes a two-pronged phosphorylation strategy to blunt the activation of cGAS.

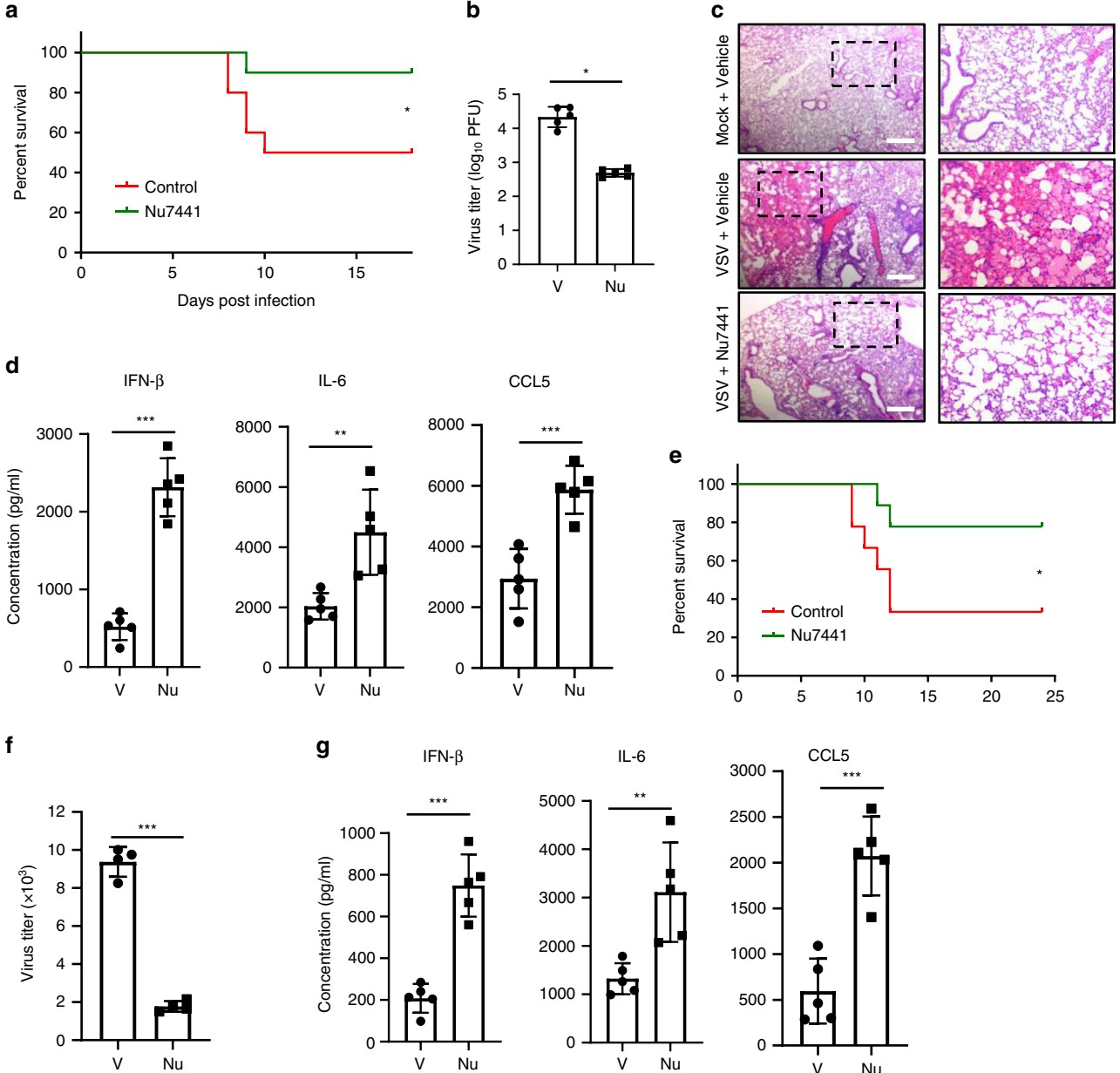

**Fig. 6 Nu7441 treatment promotes antiviral innate immunity and suppresses VSV and HSV-1 replication in vivo. a–d** Age- and gender-matched BL/6 mice were treated with 20 mg/kg Nu7441 or control through intraperitoneal injection and infected with VSV ($10^8$ PFU). **a** Mouse survival was recorded and shown as percentage over time ($n = 10$ biologically independent animals, $p = 0.046$). **b** Mice were sacrificed at 3 days post-infection, and viral titers in the spleens were quantified by plaque assays ($n = 5$ biologically independent animals, $p = 3 \times 10^{-6}$). **c** H&E-stained lung sections from the indicated mice 3 days post-infection. (Scale bar = 200 μm). **d** Blood was collected at 6 h post-infection, and the indicated cytokines in sera were determined by ELISA ($n = 5$ biologically independent animals, $p = 1 \times 10^{-5}$ (IFN-β); $p = 0.0060$ (IL-6); $p = 0.0008$ (IL-6)). **e, g** Age- and gender-matched BL/6 mice were treated with 20 mg/kg Nu7441 or control through intraperitoneal injection and infected with HSV-1 ($5 \times 10^7$ PFU). **e** Mouse survival was recorded and shown as percentage over time ($n = 9$ biologically independent animals, $p = 0.045$). **f** Mice were sacrificed at 3 days post-infection, and viral titers in the brains were quantified by plaque assays ($n = 4$ biologically independent animals, $p = 2 \times 10^{-6}$). **g** Blood was collected at 6 h post-infection, and the indicated cytokines in sera were determined by ELISA ($n = 5$ biologically independent animals, $p = 8 \times 10^{-5}$ (IFN-β); $p = 0.0060$ (IL-6); $p = 0.0004$ (IL-6)). All experiments were done at least twice, and one representative is shown. For **b, d, f, g**, data are presented as mean values ± SD. $*p < 0.05$, $**p < 0.01$, $***p < 0.005$, two-tailed Student's $t$ test (**b, d, f, g**) and log-rank test (**a, e**). Source data are provided as a Source data file.

Previously, cytoplasmic DNA-PK has been reported to be a sensor for cytosolic DNA with its kinase activity dispensable for the induction of the innate immune response[27]. Moreover, DNA-PK activates STING-independent DNA-sensing pathway in human cells[35], which was confirmed by our data that DNA-PK positively regulates HSV-1-induced innate immunity in the absence of STING. On the other hand, previous study also reveals that DNA-PK inhibition by Nu7441 in human

fibroblasts (with intact STING) led to higher DNA-induced innate immune responses[35]. Combining with our findings, we propose that DNA-PK functions for DNA sensing in cells lacking STING, while restricts cGAS signaling in cells with the presence of STING. Importantly, these two functions of DNA-PK are not mutually exclusive, and they may represent cell-type-specific regulatory roles of DNA-PK on innate immunity.

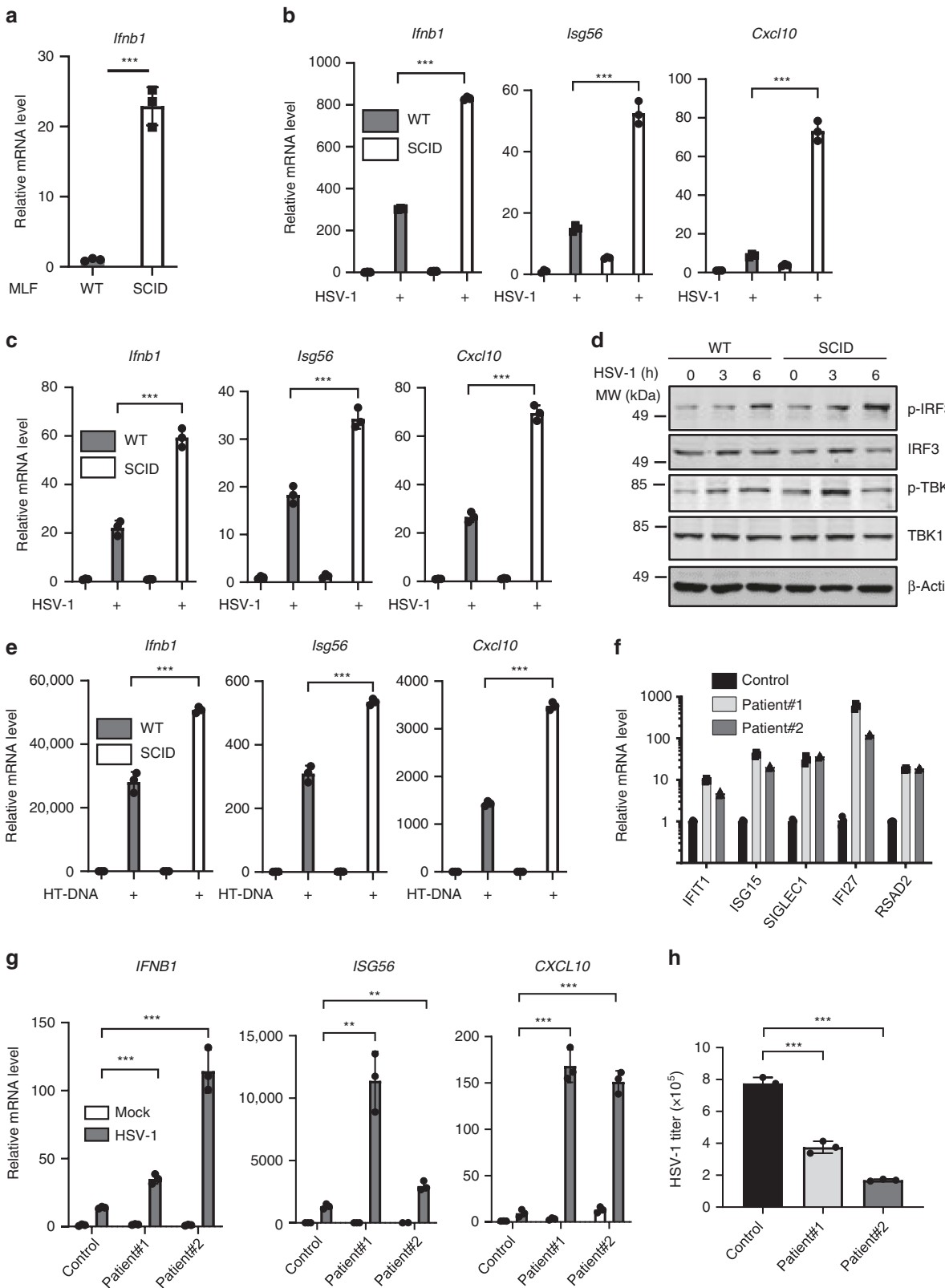

The importance of cGAS-STING signaling in autoimmunity has been highlighted by genetic studies of human patients with other autoimmune diseases[39]. These studies provide substantive evidence indicating that aberrant cGAS-STING signaling is a driver of autoimmune diseases. For example, loss-of-function mutations of cytosolic nucleases in patients lead to interferonopathies due to accumulated cytosolic DNA that activates

cGAS[40,41]. Thus cGAS-STING signaling needs to be tightly regulated to maintain immune homeostasis and cGAS has been considered a promising therapeutic target. One prominent example is that genetic KO of cGAS reverses the autoimmunity-related phenotype in a Trex1-deficient Aicardi–Goutieres Syndrome mouse model[42,43]. Recently, cGAS acetylation and inhibition by aspirin has been shown to effectively suppress

**Fig. 7 DNA-PK deficiency potentiates antiviral innate immunity in cells isolated from mice and patients. a** RNA was extracted from mouse lung fibroblasts (MLFs) generated from WT or SCID mice and the expression of *Ifnb1* was analyzed by real-time PCR. $p = 0.0002$. **b** MLFs were infected with HSV-1 (MOI = 5). Cells were harvested at 6 h post-infection, and the expression of the indicated genes was analyzed by real-time PCR. $p = 2 \times 10^{-6}$ (*Ifnb1*); $p = 8 \times 10^{-5}$ (*Isg56*); $p = 3 \times 10^{-5}$ (*Cxcl10*). **c** Bone marrow-derived macrophages (BMDMs) were infected with HSV-1 (MOI = 5). Total RNA was extracted at 6 h post-infection and the expression of the indicated genes was determined by real-time PCR. $p = 0.0002$ (*Ifnb1*); $p = 0.0006$ (*Isg56*); $p = 4 \times 10^{-5}$ (*Cxcl10*). **d** MLFs generated from WT or SCID mice were infected with HSV-1 (MOI = 5). Cells were harvested at the indicated time points, and WCLs were analyzed by immunoblotting with the indicated antibodies **e** MLFs generated from WT or SCID mice were transfected with HT-DNA (1 μg/ml). Cells were harvested at 6 h post-transfection, and the expression of the indicated genes was analyzed by real-time PCR. $p = 0.0003$ (*Ifnb1*); $p = 0.0001$ (*Isg56*); $p = 3 \times 10^{-6}$ (*Cxcl10*). **f** Total RNA was extracted from control subject or patients with DNA-PKcs mutations and the expression of the indicated genes was determined by real-time PCR[32]. **g** Fibroblasts generated from control subject or patients with DNA-PKcs mutations were infected with HSV-1 (MOI = 5). Cells were harvested at 6 h post-transfection, and the expression of the indicated genes was analyzed by real-time PCR. *IFNB1*: $p = 0.0005$ (patient#1), $p = 0.0004$ (patient#2); *ISG56*: $p = 0.0078$ (patient#1), $p = 0.0060$ (patient#2); *CXCL10*: $p = 0.0001$ (patient#1), $p = 4 \times 10^{-5}$ (patient#2). **h** Fibroblasts generated from control subject or patients with DNA-PKcs mutations were infected with HSV-1 (MOI = 0.05). Viral titer in the medium was determined by plaque assays at 24 h post-infection. $p = 0.0020$ (patient#1), $p = 1 \times 10^{-5}$ (patient#2). All experiments were done at least twice, and one representative is shown. n = 3 biologically independent samples for **a–c**, **e–h**. Data are presented as mean values ± SD. **$p < 0.01$, ***$p < 0.005$, two-tailed Student's *t* test. Source data are provided as a Source data file.

DNA-mediated autoimmunity in Aicardi–Goutieres patient cells[44]. Therefore, identifying mechanisms to prevent inappropriate activation of cGAS is crucial for the treatment of the relevant autoimmune diseases. Our study reveals that DNA-PKcs deficiency or missense mutations result in aberrant cGAS activation that leads to overactivation of innate immunity, making cGAS a promising target to treat autoimmune diseases caused by DNA-PKcs deficiency. Several inhibitors targeting cGAS and STING have been developed recently, and we propose that inhibition of cGAS-STING signaling with these inhibitors may alleviate autoimmune symptoms in these patients[45–47].

In summary, we reported that DNA-PK phosphorylates cGAS and restricts its activity. A DNA-PK inhibitor shows potent antiviral activity by enhancing antiviral innate immunity and it has the potential to be repurposed to treat infectious diseases. Fibroblasts isolated from human patients with DNA-PKcs mutations have elevated antiviral innate immunity. Our study provides an explanation for the development of autoimmune diseases in these patients and suggests that the cGAS–STING signaling axis can be a promising therapeutic target.

## Methods

**Mice and patient samples**. Six-to-8-week-old, gender-matched mice (male and female) were used for all experiments. The protocol was approved by the Institutional Animal Care and Use Committee (IACUC) of Medical Research Institute, Wuhan University. All mice were housed in a standard pathogen-free animal facility, with controlled temperature (~20 °C), humidity (40–50%), and light/dark cycle (12 h/12 h). BALB/c and BALB/c SCID mice were purchased from Hunan Silaikejingda Laboratory Animal Technology Co. Ltd (Changsha, China). C57BL/6J mice were purchased from the Jackson Laboratory. cGAS KO mice were provided by Dr. Hong-Bing Shu (Wuhan University).

A statement of informed consent was obtained from the participants. The study (fibroblast and blood sample collection and cytokine profiling of whole patient blood cells) was approved by the Medical Ethics Committee of Sud Est III (Lyon, France) and was carried out in accordance with the Declaration of Helsinki principles. Fibroblasts with DNA-PKcs mutations from two patients were provided by Dr. Alexandre Belot (University of Lyon) and Dr. Isbelle Rouvet (Hospices Civils de Lyon). The mutation of DNA-PKcs in these two patients have been described (Sanger sequencing identified a c.9185T>G (p.Leu3062Arg) homozygous missense mutation and a homozygous c.6340delGAG (p.Gly2113del) deletion in *PRKDC* in patient#1. An independent patient#2 has the same two homozygous mutations)[32]. Cytokine expression profiling of whole patient blood cells from the same two patients was determined by quantitative real-time polymerase chain reaction (qRT-PCR). The study (virus infection study of patient fibroblasts) was approved by the Medical Ethics Committee of Medical Research Institute, Wuhan University (Wuhan, China).

**Viruses**. HSV-1, VSV, and VSV carrying luciferase (VSV-Luc) or GFP were propagated using VERO cells. Virus titer was measured by standard plaque assay by using VERO cells[48]. SeV was purchased from Charles River Laboratories[48].

For VSV-GFP quantification using flow cytometry, THP-1 cells were infected with VSV-GFP (multiplicity of infection (MOI) = 0.01) for 16 h. The cells were

fixed with 10% formaldehyde for 10 min at room temperature and GFP-positive cell percentage was quantified by BD FACS Canto II. The data were analyzed by FlowJo 7.6.

**Antibodies and reagents**. Commercially available antibodies used for this study include mouse monoclonal FLAG M2 antibody (1:5000) (Sigma); rabbit polyclonal V5 antibody (1:5000) (Thermo Fisher); mouse monoclonal V5 (1:5000), β-actin (1:5000), and TBK1 antibody (1:1000) (Abcam); Phospho-TBK1 (Ser172) (1:1000), Phospho-IRF3 (Ser396) (1:1000), Ku80 (1:1000), cGAS (D1D3G) (1:1000), and STING (D2P2F) antibody (1:1000) (Cell Signaling); GAPDH (6C5) (1:1000), VDAC1 (B-6) (1:1000), and IRF3 antibody (FL-425) (1:1000) (Santa Cruz); DNA-PKcs (1:1000) and Ku70 antibody (1:1000) (NeoMarkers); and phospho-Histone H2A.X (Ser139) antibody (1:1000) (Millipore).

Rabbit polyclonal anti-cGAS pT68 and pS213 antibodies were generated by GenScript Company (NJ, USA) and used at 1:500 dilution. Rabbit polyclonal anti-MAVS antibody was generated in the laboratory and used at 1:1000 dilution[48].

Nu7441 (Selleck Chemical), Nu7026 (MedChemExpress), Herring testis (HT)-DNA (Sigma-Aldrich), poly I:C (InvivoGen), 2',3'-cGAMP (InvivoGen), streptavidin agarose (Thermo Fisher), Ni-NTA His-Bind Resin (Novagen), and Malachite Green Phosphate Detection Kit (Cell Signaling) were used according to the manufacturer's protocol. Biotin-labeled ISD-45 DNA was ordered from IDT. [α-P32]-ATP was ordered from Perkin Elmer. Lipofectamine 2000 was purchased from Invitrogen.

**Cell culture**. HEK293T and VERO cells (ATCC) were cultured in Dulbecco's modified Eagle's medium (DMEM) supplemented with 10% fetal calf serum (FCS) and penicillin–streptomycin at 37 °C. HFF cells (kindly provided by Dr. Bo Zhong, Wuhan University and Dr. Min-Hua Luo, Wuhan Institute of Virology) were cultured with DMEM supplemented with 10% FCS. THP-1 cells (ATCC) were cultured in RPMI-1640 supplemented with 10% FCS and penicillin–streptomycin at 37 °C. L929 WT and cGAS KO cells (kindly provided by Dr. James Chen, UT Southwestern Medical Center) were cultured with DMEM supplemented with 10% FCS. Primary BMDMs were differentiated and maintained in DMEM supplemented with 30% L929 conditioned medium[13]. For preparation of MLFs, mouse lungs were minced and digested in calcium- and magnesium-free Hank's Balanced Salt Solution containing 10 μg/ml type II collagenase and 20 μg/ml DNase I (Invitrogen) at 37 °C for 1 h. Cell suspension was centrifuged at $300 \times g$ for 5 min, and the cells were then plated in culture medium (1:1 [v/v] DMEM/Ham's F-12 supplemented with 10% FCS, penicillin–streptomycin, 15 mM HEPES, and 2 mM L-glutamine)[49]. Gender of the cell lines was not a consideration in this study.

**RNA extraction and qRT-PCR**. Human THP-1 monocytes, human primary fibroblasts, mouse L929 fibroblasts, mouse BMDMs, or MLFs were infected with VSV or HSV-1 (MOI = 5) or transfected with HT-DNA (1 μg/ml), unless specifically indicated otherwise. Fibroblasts from control subjects and patients #1 and #2 were infected with VSV or HSV-1 (MOI = 5) for 6 h. Cells were washed with cold phosphate-buffered saline (PBS), and total RNA was extracted using TRIzol reagent (Takara). RNA was digested with DNase I (New England Biolabs) to remove genomic DNA. The two patient blood samples were collected and frozen at −20 °C. One milliliter of blood was used to extract total RNA by a PAXgene Whole-Blood RNA Isolation Kit (PreAnalytiX, Qiagen). One microgram of total RNA was used for reverse transcription with PrimeScript Reverse Transcriptase (Clontech) according to the manufacturer's instructions. Approximately 0.5% of the cDNA was used as template in each qRT-PCR reaction with SYBR master mix (Bio-Rad) or Taqman (Thermo Fisher). Taqman probes were used to quantify IFIT1 (Hs00356631_g1), ISG15 (Hs00192713_m1), SIGLEC1 (Hs00988063_m1),

IFI27 (Hs01086370_m1), and RSAD2 (Hs01057264_m1). The relative expression of the target genes was normalized to the expression level of HPRT1 (Hs03929096_g1) or ACTB.

The primer sequences for qRT-PCR are provided in Supplementary Table.

**Luciferase reporter assay.** HEK293T cells in 24-well plates were transfected with a reporter plasmid mixture containing 50 ng of the plasmid expressing IFN-β firefly luciferase reporter and 20 ng of the plasmid expressing TK-renilla luciferase reporter. At 30 h post-transfection, cells were harvested and cell lysates were prepared. Cell lysates were used for dual luciferase assay according to the manufacturer's instructions (Promega).

**Immunoprecipitation.** Immunoprecipitation was carried out to assess protein–protein interaction[50]. Briefly, HEK293T or THP-1 cells were infected with HSV-1 or VSV (MOI = 0.5) for 16 h. Cells were harvested and lysed with NP40 buffer (50 mM Tris-HCl [pH 7.4], 150 mM NaCl, 1% NP-40, 1 mM EDTA, 5% glycerol) supplemented with a protease inhibitor cocktail (Roche). Centrifuged cell lysates were precleared with Sepharose 4B beads and incubated with 10 µl of FLAG-agarose (Sigma-Aldrich) or 0.5 µg of the indicated antibody plus 10 µl of protein G-Sepharose (GE Healthcare) at 4 °C for 4 h. Agarose beads were washed three times with lysis buffer, and precipitated proteins were released by boiling with 1× sodium dodecyl sulfate (SDS) sample buffer at 95 °C for 5 min. Precipitated proteins were resolved by SDS–polyacrylamide gel electrophoresis (PAGE) and analyzed by immunoblotting.

**Protein purification.** pET28a-hcGAS construct was kindly provided by Dr. Fanxiu Zhu (Florida State University)[5]. WT cGAS or its mutants were expressed in *Escherichia coli* BL21-DE3 cells by induction with 0.5 mM IPTG at 18 °C overnight. Cells were collected and sonicated in lysis buffer (20 mM Tris-Cl, pH 8.0, 150 mM NaCl, 5% glycerol, 10 mM imidazole, 1% Triton X-100, 0.2 mM PMSF). The supernatant was collected and incubated with Ni-NTA beads. After washing with buffer (20 mM Tris-Cl, pH 8.0, 150 mM NaCl, 5% glycerol, 20 mM imidazole), bound protein was eluted with elution buffer (20 mM Tris-Cl, pH 8.0, 150 mM NaCl, 300 mM imidazole). The proteins were dialyzed (20 mM Tris-Cl, pH 8.0, 150 mM NaCl, 10% glycerol) and used for kinase assay and in vitro enzymatic assay.

**In vitro cGAMP activity assay.** cGAS activity was assayed by using the pyrophosphatase-coupled assay[51]. One microgram of WT or mutant cGAS was mixed with 100 µM ATP, 100 µM GTP, and 0.1 U Pyrophosphatase (Thermo Fisher) in reaction buffer (20 mM Tris-Cl [pH 7.5], 150 mM NaCl, 5 mM MgCl₂, 1 mM dithiothreitol [DTT]) with or without HT-DNA (1 µg/ml). After 1 h of incubation at 25 °C, the reaction was stopped by adding an equal volume of quench solution (reaction buffer minus Mg++ plus 25 mM EDTA). Quenched solutions (25 µl) were mixed with 25 µl malachite green solution (Cell Signaling) and incubated for 15 min at RT. Absorbance at 620 nm was compared to phosphate standard to determine the concentration of phosphate in each well. Phosphate concentrations of control reactions without recombinant cGAS were subtracted from reactions containing recombinant cGAS. The measurement was repeated at least three times, and the results are presented as the enzymatic activity percentage against WT cGAS.

**In vitro kinase assay.** In vitro kinase assay was performed to assess the phosphorylation of cGAS by DNA-PK[52]. Reaction mixtures, containing 10 U of purified DNA-PK (Promega), 0.5 µg of purified cGAS or mutants, and 10 µCi of [γ32P]-ATP in 30 µl of total volume, with or without HT-DNA (1 µg/ml), were incubated at 25 °C for 30 min. Reactions were stopped with SDS/PAGE sample buffer by boiling for 5 min at 100 °C, and samples were resolved by SDS/PAGE, transferred to polyvinylidene difluoride membrane, and analyzed by autoradiography

**Stable cell line generation.** Short hairpin RNA (shRNA) targeting DNA-PKcs, Ku70, and Ku80 was kindly provided by Dr. Chengyu Liang (University of Southern California). Single guide RNA (sgRNA) targeting cGAS, STING, and MAVS was constructed into Lenti-CRISPRv2 vector. cGAS and its mutants were constructed in pcDH-CMV-EF1-Puro or pcDH-CMV-EF1-Hygro with FLAG or V5 tag.

Lentivirus was generated in 293T cells[53]. 293T, THP-1, or L929 cGAS KO cells were infected with lentivirus containing cGAS, shRNA, or sgRNA. After 36 h, 293T and THP-1 cells were selected with puromycin at 1 µg/ml and L929 CGAS KO cells were selected with puromycin at 5 µg/ml for 2 days. For the generation of 293T cGAS-V5 stable cells, 293T cells were selected with hygromycin (200 µg/ml) for 5 days. Cells were maintained in culture medium containing puromycin or hygromycin.

**Mouse infections.** Age- and gender-matched WT or cGAS KO mice were infected with HSV-1 (5 × 10⁷ plaque-forming units) or VSV (1 × 10⁸) via intraperitoneal injection for viral pathogenesis analysis. Mouse survival was monitored daily for up to 3 weeks. Sera were collected at 8 h post-infection to measure cytokine production by enzyme-linked immunosorbent assay (ELISA).

Three to five mice from each group were sacrificed at 3 days post-infection. Homogenates from the brains or spleens of infected mice were centrifuged at 12,000 rpm for 10 min at 4 °C, and supernatants were collected and used to measure viral titer by plaque assay. No mice were excluded from the quantifications, and no randomization method was used to divide the mice. The experiments were carried out non-blinded.

**Cytokine measurement.** THP-1 cells were infected with VSV (MOI = 0.05) for 16 h. The supernatant was collected and used to determine cytokine concentration.

Mice were infected with HSV-1 or VSV and sera were collected at 8 h post-infection for cytokine detection.

ELISA kit was used to determine the concentration of human IFN-β (PBL Assay Science), murine IFN-β (PBL Assay Science), CCL5 (R&D systems), and IL-6 (BD Biosciences) according to the manufacturer's instructions.

**Phosphorylation site analysis by liquid chromatography tandem mass spectrometry (LC/MS/MS).** In vitro kinase assay was performed with purified DNA-PK and cGAS in the presence of HT-DNA (1 µg/ml). Samples were resolved by SDS/PAGE and cGAS containing gel bands were treated with DTT for reduction, then iodoacetamide for alkylation, and further digested by chymotrypsin and trypsin. The digested peptide mixture was analyzed by LC/MS/MS.

The LC/MS/MS analysis of samples was carried out using a Thermo Scientific Q-Exactive hybrid Quadrupole-Orbitrap Mass Spectrometer and a Thermo Dionex UltiMate 3000 RSLCnano System. Peptide mixtures from each sample were loaded onto a peptide trap cartridge at a flow rate of 5 µl/min. The trapped peptides were eluted onto a reversed-phase PicoFrit column (New Objective, Woburn, MA) using a linear gradient of acetonitrile (3–36%) in 0.1% formic acid. The elution duration was 60 min at a flow rate of 0.3 µl/min. Eluted peptides from the PicoFrit column were ionized and sprayed into the mass spectrometer using a Nanospray Flex Ion Source ES071 (Thermo) under the following settings: spray voltage, 1.8 kV, capillary temperature, 250 °C. Other settings were empirically determined. Peptide and possible phosphorylation modification identification and protein assembly were performed on a Thermo Proteome Discoverer 1.4.1 platform. For each MS/MS data set, a single search was performed against the corresponding Uni-ProtKB/Swiss-Prot database using the SEQUEST and percolator algorithms. Carbamidomethylation (+57.021 Da) of cysteines was set as fixed modification, and Phospho/+79.966 Da (S, T, Y) were set as dynamic modifications. The minimum peptide length was specified to be five amino acids. The precursor mass tolerance was set to 15 ppm, whereas fragment mass tolerance was set to 0.05 Da. The maximum false peptide discovery rate was specified as 0.01 or 0.05. The resulting Proteome Discoverer Report contains all assembled proteins with peptides sequences, possible post-translational modifications (e.g., phosphorylation), and matched spectrum counts.

**Immunofluorescence.** THP-1 cells were mock treated or treated with Nu7441 (3 µM) for 6 h or etoposide (10 µM) for 2 h. The cells were washed with PBS and fixed for 10 min with 4% paraformaldehyde, followed by permeabilization with 1% Triton X-100 for 5 min. The samples were then blocked with 10% normal goat serum in PBS for 2 h at room temperature, followed by incubation with γ-H2AX antibodies (1:200) for 2 h at 37 °C. Slides were then washed with 0.5% NP-40–PBS and incubated with goat anti-mouse Alexa Fluor 594 (Thermo Fisher) for 1 h at room temperature. The cells were mounted with ProLong Diamond Antifade Mounting medium (Invitrogen) and observed with a Zeiss confocal microscope.

**Biotin-ISD pulldown.** The cGAS WT and mutant proteins were expressed in HEK293T cells and whole-cell lysates were mixed with biotinylated ISD (sense strand sequence 5′-TACAGATCTACTAGTGATCTATGACTGATCTGTA-CATGATCTACA-3′) at 4 °C for 4 h. The lysates were then mixed with Streptavidin beads (Thermo Fisher) for 2 h. The beads were washed three times with lysis buffer, and precipitated proteins were released by boiling with SDS sample buffer at 95 °C for 5 min. Precipitated proteins were resolved by SDS–PAGE and analyzed by immunoblotting.

**Quantification of mtDNA.** FLAG-cGAS was expressed in HEK293T cells and then pulled down by using FLAG-M2 agarose (Sigma). The binding DNA was eluted by incubating with lysis buffer (50 mM Tris, 50 mM EDTA, 1% SDS, pH 8.0) supplemented with 0.1 mg/ml Proteinase K (Qiagen) at 55 °C overnight. Then RNAse A (0.05 mg/ml) was added to the lysed samples and the samples were incubated at 37 °C for 30 min. Samples were cooled on ice before addition of pre-chilled 7.5 M ammonium acetate (Sigma) to precipitate proteins. Then the samples were centrifuged at 4000 × g for 10 min. The supernatant was mixed with 1 ml 100% isopropanol and centrifuged at 4000 × g for 10 min. The precipitated DNA was washed with 70% ethanol and resolved with 100 µl of TE buffer. The quantification of mtDNA was performed by using real-time PCR[10]. Twenty nanograms of a purified plasmid encoding for enhanced GFP (EGFP) gene was added as an internal control. The mix of endogenous DNA and EGFP plasmid was used to quantify the presence of specific DNA fragments by qPCR. Primer sets for human

mtDNA are shown below. Relative levels of the DNA molecules of interest were calculated based on the Ct values of EGFP gene amplification. The primer sequences are provided in Supplementary Table.

**Hematoxylin and eosin (H&E) staining**. Mouse lung tissue samples were fixed in 10% (vol/vol) formalin solution (Sigma) overnight. Tissue specimens were dehydrated, embedded in paraffin, and cut into 3-μm sections. Tissue sections were analyzed by standard H&E staining.

**Statistical analysis**. Data represent the mean of at least three independent experiments, and error bars denote SD unless specified otherwise. A two-tailed Student's $t$ test was used for statistical analysis except the mouse survival analysis. The mouse Kaplan–Meier survival curve was generated using GraphPad Prism 7, and the log-rank test applied to the mouse survival data was also performed in GraphPad 7. For all statistical analysis, $*p < 0.05$, $**p < 0.01$, $***p < 0.005$.

**Reporting summary**. Further information on research design is available in the Nature Research Reporting Summary linked to this article.

## Data availability

All data are included in the Supplemental Information or available from the authors upon reasonable requests. Source data are provided with this paper.

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

## Acknowledgements

We thank Dr. Bo Zhong (Wuhan University), Dr. Min-Hua Luo (Wuhan Institute of Virology), Dr. Fanxiu Zhu (Florida State University), Dr. Zhijian James Chen (UT Southwestern Medical Center), Dr. Chengyu Liang (University of Southern California), and Dr. Yanick Crow (University of Edinburgh) for reagents and data sharing. We thank Dr. Jon Hao (Poochon Scientific) for protein phosphorylation analysis. We thank Dr. Bo Zhong, Dr. Ming-Ming Hu, and Dr. Tian Xia (Wuhan University) for suggestions. J. Zhang is supported by a startup fund from Wuhan University, a grant from National Natural Science Foundation of China (31970156), and the Fundamental Research Funds for the Central Universities (2042019kf0202). P.F. is supported by grants from NIH (R35DE027556, R01DE026003, R01CA221521, R21AI134105) and NCI (CA221521).

## Author contributions

J. Zhang, P.F., and H.-B.S. designed and supervised the study; X.S., T.L., and J. Zhao performed most experiments; H.X. and C.P. performed in vitro kinase assay; J.X. and Y.G. helped with animal studies and analysis; L.Z., M.L., and Q.Y. helped with constructing multiple plasmids and animal studies; I.R. and A.B. provided patient samples and helped with the related analysis; J. Zhang, J. Zhao, and P.F. wrote the paper; all the authors approved the manuscript.

## Competing interests

The authors declare no competing interests.
