## [Peer Review File · Nature Communications]

REVIEWER COMMENTS

Reviewer #1 (Remarks to the Author):

The authors describe a role for DNA-PK in regulation of cGAS signaling. This is an interesting observation, which needs to be explored. Also, since there is a significant literature already on DNA-PK and DNA-activated immune signaling the relation between these seemingly contradicting literature needs to be addressed. In this respect, the mechanism needs to be thoroughly characterized, and the physiological relevance need to be further examined.

1. The authors need to address the recent publication from the Stetson group (PMID: 31980485). From my perspective, there seems to be contradiction, please address both experimentally and in the discussion.
2. Figure 1. THP 1 is not a physiologically relevant model to test HSV1 replication in, and also VSV I would say. This data need to be repeated in a cell system relevant for these infections.
3. Figure 2. The data on VSV and IFN are a bit confusing, in relation to the conclusion of the work. Is VSV-induced IFN responses in THP1 cells cGAS-STING dependent?
4. The authors should rigorously show that the inhibitor used work and are specific.
5. The authors rely heavily on Nu7441. The findings should be confirmed with 1-2 other inhibitors
6. Although, the authors make an attempt to provide some mechanistic insight into the MOA of DNA-PK in regulating cGAS activity, this appears somewhat superficial. It is very important to stress that there are numerous papers suggesting a positive role for DNA-PK in innate immune sensing (in addition to 31980485, for instance, Ferguson et al provided an early description of the phenomenon). So, the authors have a large body of literature that they should address. In the absence of that, I find it difficult to understand the seemingly contradicting literature.
7. Figure 6. The in vivo data should be repeated in cGAS-deficient mice. If the proposed mechanism is correct, these should be no effect of the inhibitor in that system.

Reviewer #2 (Remarks to the Author):

The work by Sun et al revealed a role of DNA-PK in the regulation of cGAS DNA sensing. The study began with a screening of small chemical library that identified a DNA-PK inhibitor Nu7447 to suppress VSV (a RNA virus) replication in THP1 cells. Further experiments suggested that Nu7447

also inhibited a DNA virus HSV replication. More detailed analyses revealed that Nu7447 potentiated VSV- and HSV-induced innate immune responses mainly through cGAS-STING pathway. The authors demonstrated that activated DNA-PK can phosphorylate cGAS on T68 and S213, and that phosphorylation of these sites diminished cGAS activity. Furthermore, they also showed that Nu7447 treatment of mice decreased viral replication and disease severity in mice. Lastly, they demonstrated that DNA-PK deficiency potentiated the host antiviral responses using murine cells from K/O mice and human cells with inborn DNA-PK deficient mutations. This is a quite comprehensive study with interesting observations. However, there are some significant discrepancies with recent literatures that should be resolved to avoid further confusions in the field.

1. DNA-PK and its components have been reported as DNA sensors by several groups. Some of those have been cited in the paper. Dr. Stetson's lab recently published that DNA-PK can sense DNA in a STING-independent manner in human cells but not in murine cells. They claimed that Nu7447 inhibited DNA-induced innate immune responses. These and other discrepancies should be carefully addressed. I noticed that Stetson's study used transfected DNA while the current study used VSV and HSV. I recommend to repeat some critical experiments using transfected DNA (maybe RNA as a control) as stimulant of innate immune response VS viral infections, and comparing the effects of Nu7447 and DNA-PK deficiency. Also, can transfection of DNA (as comparison to viral infections in Figure 4a) induce interaction between cGAS and DNA-PK?

2. An important question is how DNA-PK is activated by viruses especially RNA viruses such VSV? It is interesting that VSV and Sendai viruses behaved differently. What makes them behave differently? I noticed that they were used at very different dose or MOI. Curiously, VSV as used at MOI of 0.01 in most experiments (although there were some discrepancies in the Figure legends, Methods, and Text) which means that less than 1% of cells were infected. Is the low MOI critical for the observed differences throughout the work?

3. The Nu7447 itself seems to have no apparent effect on innate immune responses. Does it increase the responses induced by DDR-inducing drugs?

4. Do T68 and S213 match to the known DNA-PK phosphorylation motif? Can the site specific phosphorylation be detected in THP1 cells during viral infections?

5. Figure 5D, it's less convincing if they were not on the same blot. Can the original blot be shown?

RESPONSE to REVIEWER COMMENTS

Reviewer #1 (Remarks to the Author):

The authors describe a role for DNA-PK in regulation of cGAS signaling. This is an interesting observation, which needs to be explored. Also, since there is a significant literature already on DNA-PK and DNA-activated immune signaling the relation between these seemingly contradicting literature needs to be addressed. In this respect, the mechanism needs to be thoroughly characterized, and the physiological relevance need to be further examined.

Response: We thank the reviewer for her/his positive and constructive comments on our manuscript. We address her/his concerns as detailed below. Whenever possible, experiments were performed to provide answers to the reviewer's questions.

1. *The authors need to address the recent publication from the Stetson group (PMID: 31980485). From my perspective, there seems to be contradiction, please address both experimentally and in the discussion.*

Answer: Thanks very much for pointing out this. A recent study from Dr. Daniel Stetson's group (Burleigh et al., Science Immunology, 2020) reports that human DNA-PK activates **STING-independent** DNA sensing pathway in human cells. In other words, their data reveal that DNA-PK is required for DNA sensing **in the absence of STING**. In contrast, we report here that DNA-PK restricts cGAS-STING signaling **in the presence of STING**. These two conclusions are not mutually exclusive, and in fact, data from Dr. Stetson's paper (shown below) also support that DNA-PK inhibition by Nu7441 in human fibroblast (with intact STING) led to higher DNA-induced innate immune response, which is consistent with our conclusion.

(Burleigh et al., Science Immunology, 2020)

To further address this question, we repeated Nu7441 treatment with THP-1 STING-KO cells and found that HT-DNA-induced innate immune responses are strictly dependent on STING, while Nu7441 failed to regulate HT-DNA-induced innate immune responses in STING-KO THP-1 cells.

In contrast to HT-DNA, we found that HSV-1 induced innate immune responses in STING-KO THP-1 cells. Importantly, Nu7441 treatment **reduced** HSV-1-induced innate immune responses in STING-KO cells, which is consistent with Dr. Stetson's conclusion that DNA-PK positively regulates HSV-1-induced innate immunity in the absence of STING.

We also performed VSV stimulation in THP-1 STING-KO cells and found that DNA-PK inhibition failed to enhance VSV-induced innate immune responses, which is consistent with our conclusion that cGAS-STING signaling is required for DNA-PK to limit IFN response and promote VSV replication.

We have incorporated these figures in our manuscript (new Supplementary Fig. 3g-i) and discussed the difference between our study and Dr. Stetson's study (line #440-447).

2. Figure 1. THP 1 is not a physiologically relevant model to test HSV1 replication in, and also VSV I would say. This data need to be repeated in a cell system relevant for these infections.

Answer: We appreciate the insightful suggestions from the reviewer. We now include

fibroblasts (HFF for HSV-1 and L929 for VSV) to repeat viral replication study with Nu7441 treatment and obtained similar results (new Fig. 1d and Supplementary Fig. 1c).

We further confirmed that HSV-1 and VSV induced stronger antiviral innate immune responses with Nu7441 treatment using HFF and L929, respectively (new Fig. 2b and Supplementary Fig. 2a).

3. Figure 2. The data on VSV and IFN are a bit confusing, in relation to the conclusion of the work. Is VSV-induced IFN responses in THP1 cells cGAS-STING dependent?

Answer: We thank the reviewer for the instructive comments.

Our data suggested that cGAS and STING KO eliminated the effect of Nu7441 on VSV replication, suggesting that cGAS-STING pathway contributes to antagonize VSV (Figure 3b). To answer the reviewer's question, we have infected cGAS and STING KO THP-1 cells with VSV, and found that the induction of *IFNB1* and *CXCL10* was partly dependent on cGAS and STING (new Supplementary Fig. 3d).

Based on our data, we conclude that VSV infection leads to the release of mitochondrial DNA (mtDNA) that promotes the activation of cGAS-STING signaling. Previous studies indicate that Dengue virus triggers cGAS-mediated antiviral immunity that is dependent on mitochondrial DNA (Aguirre et al., Nature Microbiology, 2017). We provide evidence here to show that VSV infection promotes the release of mtDNA which binds to cGAS in VSV-infected cells (Fig. 3d,e and Supplementary Fig. 3a). (Please also refer to the answer to question #2 of the second reviewer).

4. The authors should rigorously show that the inhibitor used work and are specific.

Answer: We thank the reviewer for pointing out this. We addressed the efficiency and specificity of the inhibitor Nu7441 by three sets of experiments.

First, we showed that VSV and HSV-1 infection led to the activation of DNA-PK (DNA-PKcs pS2056 as an activation marker) and Nu7441 treatment abolished the activation of DNA-PK. These data indicate that the inhibitor Nu7441 efficiently inhibits the activity of DNA-PK (new Supplementary Fig. 2b).

Second, we knocked down the expression of DNA-PKcs in THP-1 cells using shRNA and found that Nu7441 no longer increased the innate immune responses induced by HSV-1 and HT-DNA. These data demonstrate that the inhibitor Nu7441 specifically inhibits DNA-PK to enhance antiviral innate immunity (new Supplementary Fig. 2c and 3f).

Third, we got another widely used DNA-PK inhibitor Nu7026, as suggested by the reviewer in Question #5, and confirmed its immune stimulatory effect (please refer to the figures shown in Question #5).

5. The authors rely heavily on Nu7441. The findings should be confirmed with 1-2 other inhibitors

Answer: We appreciate the reviewer for the valuable suggestion. We treated THP-1 cells with another widely used DNA-PK inhibitor Nu7026 and confirmed its immune stimulatory effect (new Fig. 2e and 2f).

6. Although, the authors make an attempt to provide some mechanistic insight into the MOA of DNA-PK in regulating cGAS activity, this appears somewhat superficial. It is very important to stress that there are numerous papers suggesting a positive role for DNA-PK in innate immune sensing (in addition to 31980485, for instance, Ferguson et al provided an early description of the phenomenon). So, the authors have a large body of literature that they should address. In the absence of that, I find it difficult to understand the seemingly contradicting literature.

Answer: We appreciate the reviewer for the valuable comments.

Previously, DNA-PK has been reported as a sensor for cytoplasmic DNA (Ferguson et al., Elife, 2012). Later, cGAS is widely accepted as a critical cytoplasmic DNA sensor. It is speculated that DNA-PK may function as a DNA sensor under specific situations. Indeed, a very recent study from Dr. Daniel Stetson's group reports that human DNA-PK activates **STING-independent** DNA sensing pathway, and they show that DNA-PK is required for DNA sensing **in the absence of STING**. Nearly all data from Dr. Stetson's study used STING-deficient cells, however, one piece of their data using STING-expressing fibroblast supports our conclusion that DNA-PK suppresses cGAS-STING signaling in the presence of STING (please also refer to the answer to Question #1).

We conclude that DNA-PK plays distinct roles with or without STING. In Dr. Stetson's study, DNA-PK positively regulates cytoplasmic DNA-sensing without STING. In our study, DNA-PK suppresses cGAS-STING signaling in the presence of STING.

We have cited these papers and discussed the difference between our study and these studies as suggested by the reviewer (line #437-448).

7. Figure 6. The *in vivo* data should be repeated in cGAS-deficient mice. If the proposed mechanism is correct, these should be no effect of the inhibitor in that system.

Answer: We performed infection experiments with cGAS-KO mice and demonstrated that the inhibitor Nu7441 did not show any immunostimulatory effects in cGAS-

deficient mice (new Supplementary Fig. 6).

Reviewer #2 (Remarks to the Author):

The work by Sun et al revealed a role of DNA-PK in the regulation of cGAS DNA sensing. The study began with a screening of small chemical library that identified a DNA-PK inhibitor Nu7447 to suppress VSV (a RNA virus) replication in THP1 cells. Further experiments suggested that Nu7447 also inhibited a DNA virus HSV replication. More detailed analyses revealed that Nu7447 potentiated VSV- and HSV-induced innate immune responses mainly through cGAS-STING pathway. The authors demonstrated that activated DNA-PK can phosphorylate cGAS on T68 and S213, and that phosphorylation of these sites diminished cGAS activity. Furthermore, they also showed that Nu7447 treatment of mice decreased viral replication and disease severity in mice. Lastly, they demonstrated that DNA-PK deficiency potentiated the host antiviral responses using murine cells from K/O mice and human cells with inborn DNA-PK deficient mutations. The is a quite comprehensive study with interesting observations. However, there are some significant discrepancies with recent literatures that should be resolved to avoid further confusions in the field.

Response: We thank the reviewer for her/his positive and constructive comments. We have addressed all concerns from the reviewer to our best and hope our revision could answer the reviewer's questions.

1. DNA-PK and its components have been reported as DNA sensors by several groups. Some of those have been cited in the paper. Dr. Stetson's lab recently published that DNA-PK can sense DNA in a STING- independent manner in human cells but not in murine cells. They claimed that Nu7447 inhibited DNA-induced innate immune responses. These and other discrepancies should be carefully addressed. I noticed that Stetson' study used transfected DNA while the current study used VSV and HSV. I recomend to repeat some critical experiments using transfected DNA (maybe RNA as a control) as stimulant of innate immune response VS viral infections, and comparing the effects of Nu7447 and DNA-PK deficiency. Also, can transfection of DNA (as comparison to viral infections in Figure 4a) induce interaction between cGAS and DNA-PK?

Answer: We thank the reviewer for pointing out this.

Previously, DNA-PK has been reported as a sensor for cytoplasmic DNA (Ferguson et al., Elife, 2012). Later, cGAS is widely accepted as a critical cytoplasmic DNA sensor. We think that DNA-PK may function as a DNA sensor under specific situations. In the recent study from Dr. Daniel Stetson's group, they report that human DNA-PK activates **STING-independent** DNA sensing pathway, and DNA-PK is required for DNA sensing **in the absence of STING**. Interestingly, one piece of data from Dr. Stetson's paper indicates that DNA-PK inhibition by Nu7441 in STING-expressing fibroblasts enhances DNA-induced innate immune responses, which supports our conclusion.

(Burleigh et al., Science Immunology, 2020)

We fully take the reviewer’s suggestions and performed new experiments with HT-DNA and poly [I:C] stimulation. We found that Nu7441 increased HT-DNA-induced innate immune responses, however, the treatment didn’t affect poly [I:C]-induced innate immune responses, which is consistent with our conclusion that DNA-PK restricts cGAS-STING signaling (new Fig. 3g and h).

To further resolve the discrepancies between our study and Dr. Stetson’s study, we also performed several experiments using THP-1 STING-KO cells and found that HT-DNA-induced innate immune response is strictly dependent on STING (new Supplementary Fig. 3g).

In contrast, we found that HSV-1 induced the expression of *IFNβ1* and *CXCL10* in STING-KO cells and the induction was **reduced** by Nu7441 treatment, which is consistent with Dr. Stetson’s conclusion (new Supplementary Fig. 3h).

Together, we conclude that DNA-PK inhibits cGAS-STING signaling in the presence of STING while DNA-PK contributes to HSV-1-induced innate immune responses **if STING is depleted**. We also discussed the difference between our study and Dr. Stetson's study (line #437-448). (Please also refer to the answer to question #1 of the first reviewer).

Additionally, we did co-immunoprecipitation analysis with HT-DNA stimulation to demonstrate that HT-DNA stimulation can induce the interaction between cGAS and DNA-PK (new Supplementary Fig. 4c).

2. An important question is how DNA-PK is activated by viruses especially RNA viruses such VSV? It is interesting that VSV and Sendai viruses behaved differently. What make them behave differently? I noticed that they were used at very different dose or MOI. Curiously, VSV as used at MOI of 0.01 in most experiments (although there were some discrepancies in the Figure legends, Methods, and Text) which means that less than 1% of cells were infected. Is the low MOI critical for the observed differences throughout the work?

Answer: We thank the reviewer for the instructive questions.

VSV infection induced the dimerization and activation of cGAS while SeV could not (Supplementary Fig. 3e). Our data further indicate that cGAS associates with mitochondrial DNA in VSV-infected cells (Fig. 3e), suggesting that VSV infection leads to the release of mtDNA into cytoplasm. To corroborate these results, we performed new experiments to extract the cytoplasmic DNA and quantify cytoplasmic mtDNA. The results indicate that VSV infection led to the leak of mtDNA into cytoplasm which was not observed in SeV infection (new Fig. 3d), consistent with our data showing that VSV, not SeV, activates cGAS-STING signaling.

We conclude that VSV replication may cause more severe damage to mitochondria thus lead to the release of mtDNA. Alternatively, VSV replication may positively regulate mtDNA release.

We thank the reviewer for pointing out the MOI discrepancies. When we stimulated cells to detect innate immune response, we chose to use higher MOI (i.e., MOI=5) to synchronize the infection. When we performed multiple viral grow curve, we chose low MOI (i.e., MOI=0.01) to start with a low infection percentage. Per the reviewer's suggestion, we have clarified these MOIs in figure legends and methods. Furthermore, to address the reviewer's concerns, we use both low MOI and high MOI infection and found that they both led to cytoplasmic mtDNA increase (new Supplementary Fig. 3a). We conclude that VSV infection can activate cGAS-STING signaling regardless of the initial MOI in the experiment.

3. *The Nu7447 itself seems to have no apparent effect on innate immune responses. Does it increase the responses induced by DDR-inducing drugs?*

Answer: We thank the reviewer for the insightful suggestions. To answer this question, we treated THP-1 cells with etoposide to induce DDR and found that Nu7441 treatment further increased innate immune responses induce by etoposide (new Supplementary Fig. 2i).

4. Do T68 and S213 match to the known DNA-PK phosphorylation motif? Can the site specific phosphorylation be detected in THP1 cells during viral infections?

Answer: It's been reported that DNA-PK phosphorylates many substrates on Serine or Threonine that are followed by glutamine *in vitro*, i.e., SQ/TQ motifs (Douglas et al., Molecular and Cellular Biology, 2007). T68 matches to the TQ motif, while S213 is followed by Y which seems to be a non-traditional substrate for DNA-PK. However, we provide substantive evidence to demonstrate that both sites can be phosphorylated by DNA-PK *in vitro* and in cells.

Per the reviewer's suggestion, we immunoprecipitated endogenous cGAS in THP1 cells and demonstrate that both sites are phosphorylated during HSV-1 and VSV infection (new Supplementary Fig. 4g).

5. Figure 5D, it's less convincing if they were not on the same blot. Can the original blot be shown?

Answer: We would like to thank the reviewer to point out this. We repeated the experiments and replaced the original blots (Fig. 5d) and the uncropped versions of blots are provided in source data.

REVIEWERS' COMMENTS

Reviewer #1 (Remarks to the Author):

Although I was initially rather critical, the authors have largely convinced me now, and have done a great job in revision. I have no further comments, and hope to see how the work will do in the literature.

Reviewer #2 (Remarks to the Author):

The authors have made substantial efforts to address the critiques and I have no more comments.

REVIEWERS' COMMENTS

Reviewer #1 (Remarks to the Author):

Although I was initially rather critical, the authors have largely convinced me now, and have done a great job in revision. I have no further comments, and hope to see how the work will do in the literature.

Response: We sincerely appreciate the reviewer for her/his valuable advice during the review process.

Reviewer #2 (Remarks to the Author):

The authors have made substantial efforts to address the critiques and I have no more comments.

Response: We sincerely appreciate the reviewer for her/his valuable advice during the review process.